# ChinaSoyArea10m: a dataset of soybean planting areas with a spatial resolution of 10 m across China from 2017 to 2021

Qinghang Mei[1,2,3], Zhao Zhang[1,2], Jichong Han[1,2,4], Jie Song[1,2,4], Jinwei Dong[5,6], Huaqing Wu[1,2,3], Jialu Xu[1,2], Fulu Tao[5,6]

[1] Joint International Research Laboratory of Catastrophe Simulation and Systemic Risk Governance,Beijing Normal University, Zhuhai 519087, China

[2] School of National Safety and Emergency Management, Beijing Normal University, Beijing 100875 / Zhuhai 519087, China

[3] Faculty of Geographical Science, Beijing Normal University, Beijing 100875, China

[4] School of Systems Science, Beijing Normal University, Beijing 100875, China

[5] Key Laboratory of Land Surface Pattern and Simulation, Institute of Geographical Sciences and Natural Resources Research, Chinese Academy of Sciences, Beijing, 100101, China

[6] College of Resources and Environment, University of Chinese Academy of Sciences, Beijing 100049, China

*Correspondence to*: Zhao Zhang (zhangzhao@bnu.edu.cn)

**Abstract**

Soybean, an essential food crop, has witnessed a steady rise in demand in recent years. There is a lack of high-resolution annual maps depicting soybean planting areas in China, despite China being the world's largest consumer and fourth largest producer of soybeans. To address this gap, we developed a novel Regional Adaptation Spectra-Phenology Integration method (RASP) based on Sentinel-2 remote sensing images from the Google Earth Engine (GEE) platform. We utilized various auxiliary data (e.g., cropland layer, detailed phenology observations) to select the specific spectra and indices that differentiate soybeans most effectively from other crops across various regions. These features were then input for an unsupervised classifier (K-means), and the most likely type was determined by a cluster assignment method based on dynamic time warping (DTW). For the first time, we generated a dataset of soybean planting areas across China, with a high spatial resolution of 10 meters, spanning from 2017 to 2021 (ChinaSoyArea10m). The $R^2$ values between the mapping results and the census data at both county- and prefecture-level were consistently around 0.85 in 2017-2020. Moreover, the overall accuracy of mapping results at the field level in 2017, 2018, and 2019 were 77.08%, 85.16% and 86.77%, respectively. Consistency with census data was improved at the county level ($R^2$ increased from 0.53 to 0.84), compared to the existing 10-m crop-type maps in Northeast China (Crop Data Layer, CDL) based on field samples and supervised classification methods. ChinaSoyArea10m is spatially consistent well with the two existing datasets (CDL and GLAD maize-soybean map). ChinaSoyArea10m provides important information for sustainable soybean production and management, as well as agricultural system modeling and optimization. ChinaSoyArea10m can be downloaded from an open-data repository (DOI: https://zenodo.org/doi/10.5281/zenodo.10071426, Mei et al., 2023).

**1 Introduction**

Soybean, one of the most important crops around the world, plays an important role in diet and livestock breeding (Hartman et al., 2011). As the global demand for protein and meat increases, China's demand for soybeans has been keeping rising nowadays. In the past decade, China has averagely accounted for over 30% of the world's total soybean consumption (Liu and Fan, 2021). Despite being the fourth-largest

producer of soybeans after Brazil, the United States, and Argentina, China's self-sufficiency rate is low
(FAOSTAT, 2023; Wang et al., 2023). Given the rapid growth of demand and the shortages of domestic
supply due to lower yield and self-sufficiency, mapping soybean planting areas across China is crucial
for sustainable soybean production and management (Cui and Shoemaker, 2018; Liu et al., 2021).
Soybean planting area in some regions of China was mapped in previous studies (You et al., 2021;
Huang et al., 2022; Chen et al., 2023), but long-term soybean maps over all major producing areas in
China have not been available. A decision tree method based on phenological and near-infrared
reflectance differences was applied in the state of Parana in Brazil to produce corn-soybean maps with a
resolution of 500 m (Zhong et al., 2016). However, this study was limited to one state and a simple
planting pattern (including soybeans and corn only) at a medium resolution. The field size in China is
generally small, and 500 m-resolution maps will inevitably bring pixel mixing problem (Lowder et al.,
2016). More recently, 20-year soybean-corn maps with 30 m resolution across the US Midwest have been
generated by collecting a large number of samples and using green chlorophyll vegetation index (GCVI)
time series features, which is a large-scale, high-precision soybean mapping attempt (Wang et al., 2020).
Similarly, high-precision soybean maps in China were also made by collecting major crop samples and
utilizing spectral reflectance and vegetation indexes characteristics, for 2017-2019 in Northeast China
(You et al., 2021). Some studies have utilized unique canopy water content and chlorophyll content to
produce soybean maps in the three provinces of Northeast China from 2017 to 2021 (Huang et al., 2022).
Other studies made laudable efforts to craft a comprehensive national maize-soybean map for China in
2019 by combining field data and regression estimators (Li et al., 2023). However, these studies were
confined in some degrees because of the specific region or a single year, despite prior attempts to
accurately map soybean cultivation areas. Long-term annual soybean maps over mainly planting areas
in China with a higher spatial resolution have not been available so far.
Mapping crops by remote sensing can be categorized into four methods : 1) supervision classification
based on a large number of field samples or high quality training labels (Song et al., 2017; You et al.,
2021; Shangguan et al., 2022; Li et al., 2023); 2) developing some composite indexes based on the feature
bands and determining the binary classification using appropriate thresholds (Huang et al., 2022; Chen
et al., 2023; Zhou et al., 2023); 3) threshold segmentation based on prior knowledge such as phenology
or spectra (Zhong et al., 2016); 4) combining unsupervised classification with cluster assignment (Wang
et al., 2019; You et al., 2023). Supervision classification methods relied on ground samples heavily, while
the 2nd and 3rd methods are both based on reliable and accurate thresholds. However, mapping soybean
by these methods was mainly applied in small areas, very few covering over a larger region. Because of
sufficient field samples, supervision classification can achieve maps with a higher accuracy, which is
relatively mature method used widely. However, collecting sufficient field samples is extremely time,
money, and labor consumed, and unsuitable for long-term years over larger areas (Luo et al., 2022).
Furthermore, the threshold-based methods (the 2nd and 3rd) have been applied into large areas, however,
determining the thresholds will inevitably bring significant uncertainty, especially for the areas with high
heterogeneity in climate, environment, and planting patterns. Thus, these methods show low
reproducibility, further hindering their application across diverse geographic areas. As for mapping
soybean, it is still a big challenge due to their similar growth characteristics with many other summer
crops (Wang et al., 2020; Di Tommaso et al., 2021). The thresholds that work well in some areas did not
perform well in other areas (Graesser and Ramankutty, 2017; Guo et al., 2018). These limitations restrict
accurate soybean maps available, especially over large regions in China. Given the challenges of
collecting sufficient field samples over larger region and the limited adaptability to environmental
variations of threshold-based method, previous researches have yet to achieve multi-year, high-resolution
soybean maps nationwide.
Along this line, the adaptive classification approach tailored to distinct areas, i.e., method (4), is a
highly effective for accurately mapping crops over a larger region. Such unsupervised classification can
effectively address the above issues such as insufficient samples and limited spatial scalability by training
classifiers separately in different areas (Ma et al., 2020; Wang et al., 2022). Remarkable successes have
been achieved when applying the approach into the United States in mapping soybean and maize (Wang
et al., 2019). Due to the different climatic and environmental conditions, together with huge differences
in cultivating patterns over various areas, crop phenological information has become an important
reference for crop classification. For example, the phenological observations at the agricultural
meteorological stations were employed as a reference to detect the critical phenological dates of pixels
through inflexion- and threshold-based methods, thereby generating planting areas for three major crops
in China with $R^2$ greater than 0.8 compared to county statistics (Luo et al., 2020). The time-weighted
dynamic time warping method based on the similarity of phenological curves of Normalized Difference
Vegetation Index (NDVI) has successfully estimated the planting area of maize in China, with provincial
averages for producer's and user's accuracies at 0.76 and 0.82, respectively (Shen et al., 2022).
Phenological-based Vertical transmit Horizontal receive (VH) polarized time series accurately captured
temporal characteristics of soybeans, thus were used for an unsupervised classifier to map the seasonal
soybeans, achieving an overall accuracy over 80% in Ujjain district (Kumari et al., 2019). By integrating
unsupervised classification's regional scalability with specific local soybean growth signs from
phenological data, we fully leverage soybean's characteristic spectra and vegetation indices during key
growth periods across different areas. Through training the local unsupervised classifier to accommodate
the crop growth variability across regions, and avoiding extensive jobs on collecting samples, the
approach provides an effective solution for regional adaptive large-area crop mapping.
The main objectives of this study are: 1) to develop a novel framework to map soybean planting area
over a larger region; 2) to test the generalization ability of the framework and assess the accuracy of maps
at different levels; and 3) to provide a new data product of soybean planting area across mainly planting
areas in China, for multi-years with a high spatial resolution.
**2 Materials and methods**
**2.1 Study area**
We selected 14 major soybean producing provinces (including Chongqing Municipality) as study area,
which cover over 90% of the total planting area in China (National Bureau of Statistics of China, 2023)
(Fig. 1). The soybean planting areas were classified into four agro-ecological zones (AEZs) based on
their diverse geographical environment and planting habits, including Northeast single cropping eco-
region (NE, Zone I), Huang-Huai-Hai double cropping eco-region (HH, Zone II), Middle-Lower Yangtze
River double cropping eco-region (MLY, Zone III) and Southwest double cropping eco-region (SW, Zone
IV) (Wang and Gai, 2002). In particular, Zone I and Zone II are the main soybean producer in China,
accounting for more than 70% of the national soybean planting area.

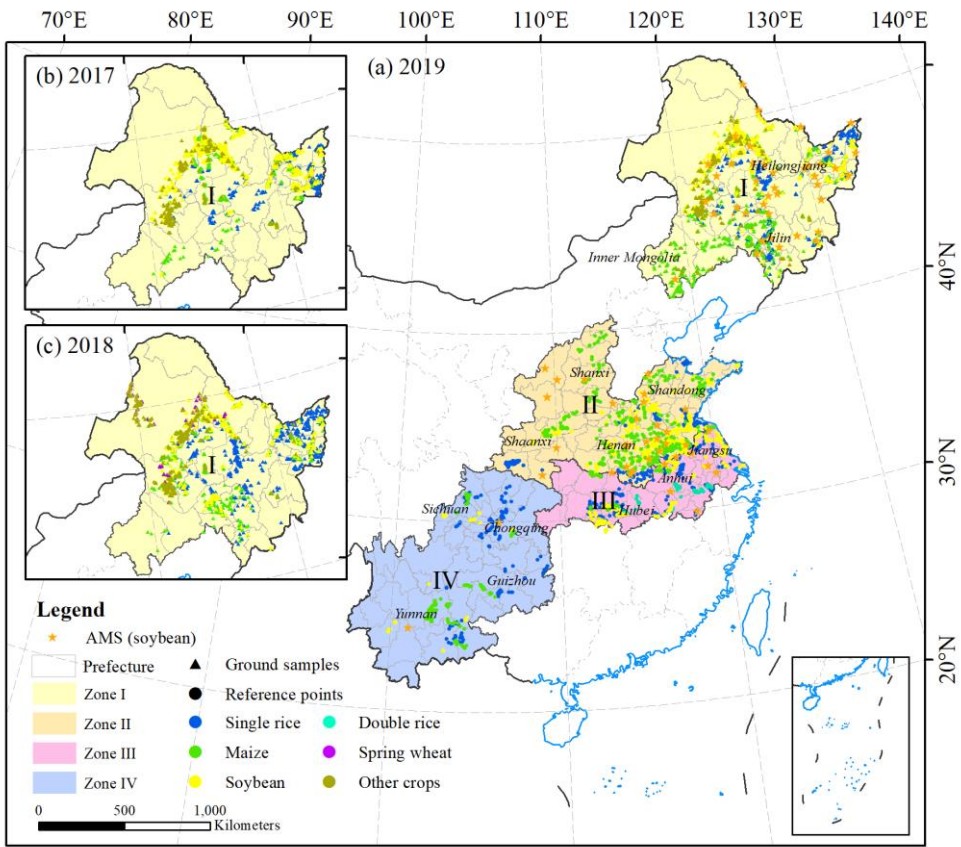


**Figure 1. The study area including 14 provinces (including Chongqing Municipality) and spatial distribution**

**of ground samples and reference points across China in (a) 2019, (b) 2017, and (c) 2018. The 14 provinces**

**include Heilongjiang, eastern Inner Mongolia, Anhui, Henan, eastern Sichuan, Jilin, Hubei, Guizhou, Jiangsu,**

**Yunnan, Shandong, Shaanxi, Shanxi, and Chongqing. Stars, triangles, and dots represent the locations of**

**soybean agricultural meteorological stations (AMSs), ground samples, and reference points, respectively.**

**2.2 Data**

**2.2.1 Remote sensing data**

We used Sentinel-2A/B Multi-Spectral Instrument (MSI) Level-1C top-of-atmosphere (TOA) reflectance

data during 2017-2021 (https://developers.google.com/earth-engine/datasets/catalog/COPERNICUS_S2,

last access: September 2023). Because of the longer-term coverage of Sentinel-2 Level-1C TOA

reflectance data, and the nearly identical spectral profile time series extracted from both products, we opt

to use L1C products instead of L2A, considering that TOA images fully meet the crop classification

requirements (You and Dong, 2020; Han et al., 2021; Luo et al., 2022). Sentinel-2 sensors provide

observations in 13 spectral bands at 10 m or 20 m resolution. The red-edge bands and shortwave infrared

bands equipped with sentinel-2 play a great role in enhancing the accuracy of crop classification (Luo et
al., 2021; Marshall et al., 2022). In addition, the S2 cloud probability dataset provided by the official can
identify cloud pollution areas and be used as cloud removal processing.

**2.2.2 In-situ phenological observations**

The soybean phenology observations in study area from 2017 to 2020 were obtained from 76 agricultural
meteorological stations (AMSs) governed by the CMA (https://data.cma.cn/, last access: May 2022).
Phenology information of each AMS is observed on alternate days or once a day, and key phenological
events such as sowing, emergence, three-true-leaves, branching, flowering, podding, full-seeding, and
maturity are noted by technicians to ensure accuracy. We defined the period from sowing to flowering as
the vegetative growth period (VGP), and the period from flowering to maturity as the reproductive
growth period (RGP) of soybeans (Gong et al., 2021). In cases of missing observation for a specific year,
we inserted the average of two closest observations before and after the year. For instance, if there was
missing data of flowering date in 2017, we filled it with the average of flowering records in 2016 and
2018 at the same station.

**2.2.3 Cropland data**

GLAD cropland product with a 30-m resolution in China was used as cropland masks
(https://glad.umd.edu/dataset/croplands, last access: September 2023) (Potapov et al., 2022). The crop
layer was conducted every four years from 2000 to 2019. We used the file for the 2016-2019 interval
which is closest to the study years. GLAD's overall accuracy of pixel-wise validation is 0.88 in China,
consistent well with the census data. The accuracy of the product is higher than that of similar products,
making it a reliable for crop mapping (Zhang et al., 2022).

**2.2.4 Census data and ground samples**

To determine the number of clusters at prefecture-level and validate the accuracy of the soybean maps at
county (2017-2018) or prefecture (2019-2020) level, we utilized agricultural census data obtained from
the statistical yearbook of each county or province by accessing National Bureau of Statistics of China
(http://www.stats.gov.cn/, last accessed: June 2023).
We used both ground samples and reference points based on available datasets to determine soybean
standard curves and assess the reliability of the soybean maps (Fig. 1). All points were randomly divided
in a 3:7 ratio for standard curve calculation and accuracy validation, respectively (Dong et al., 2020). We
collected ground samples from field surveys from 2017 to 2019 in Heilongjiang (HLJ), Inner Mongolia
(NMG), Anhui (AH), Henan (HN), and Jilin (JL), which account for more than 70% of the country's total
soybean planting area (Table 1). Crop types (soybean, maize, rice, wheat, others) and other land cover
types were recorded. To ensure the impartiality of verification results, we only selected crop samples for
validation. In provinces without ground samples, we manually selected reference points on large soybean
plots based on GLAD (https://glad.earthengine.app/view/china-crop-map, last access: March 2024)
soybean layer. The criterions selected are: (1) located in large plots; (2) false color composite image (R:
NIR, G: SWIR2, B: SWIR1) at the peak of growing season (Song et al., 2017; You and Dong, 2020); (3)
phenological characteristics similar to local observations. Additionally, the reference points of maize,
single-cropping rice and double-cropping rice in 2019 were selected based on GLAD maize layer, high
resolution single-season rice map (https://doi.org/10.57760/sciencedb.06963, last access: March 2024),
and double-season rice map (https://doi.org/10.12199/nesdc.ecodb.rs.2022.012, last access: March 2024)
with the same principle to explore the spectral characteristics of crops in each sub-zone of the studied
areas. The overall accuracy of all available maps in 2019 is above 85% (Pan et al., 2021; Li et al., 2023;
Shen et al., 2023).

**Table 1. Summary of ground samples for validation.**

|  |  | HLJ | NMG | AH | HN | JL |
|---|---|---|---|---|---|---|
| 2017 | Soybean | 1013 | 451 | - | - | 0 |
|  | Maize | 1061 | 146 | - | - | 11 |
|  | Rice | 513 | 38 | - | - | 13 |
|  | Other crops | 124 | 459 | - | - | 0 |
| 2018 | Soybean | 525 | 746 | 72 | 15 | 117 |
|  | Maize | 764 | 479 | 73 | 20 | 217 |
|  | Rice | 587 | 42 | 0 | 0 | 71 |
|  | Wheat | 10 | 141 | 0 | 0 | 0 |
|  | Other crops | 70 | 1069 | 0 | 0 | 0 |
| 2019 | Soybean | 901 | 562 | 51 | - | 26 |
|  | Maize | 468 | 463 | 53 | - | 197 |
|  | Rice | 392 | 36 | 0 | - | 148 |
|  | Other crops | 62 | 445 | 0 | - | 36 |

**2.2.5 Existing products**

We utilized the crop map CDL of Northeast China from 2017 to 2019
(https://figshare.com/articles/figure/The_10-m_crop_type_maps_in_Northeast_China_during_2017-
2019/13090442, last access: September 2023) for consistency comparison with census data, and the
2019 GLAD maize-soybean map as a reference for spatial detail comparison with ChinaSoyArea10m.
CDL is a 10m resolution crop map dataset of Northeast China from 2017 to 2019 that was created
using Sentinel-2 key spectral bands and vegetation indices, multi-year field samples, and random forest
classifiers (You et al., 2021). The maps include three crop types: rice, maize, and soybeans. The GLAD
maize-soybean Map is a national classification map for 2019 that was produced using random forests,
based on field surveys and area estimates (Li et al., 2023). The agreement ($R^2$) between GLAD and the
statistics is higher than 0.9, and the overall mapping accuracy is greater than 90%, making it a reliable
reference for comparing spatial details. We extracted the soybean layers from all the existing products.

**2.3 Methods**

Mapping soybean consists of three main steps (Fig.2): data processing, soybean mapping, and accuracy
assessment. It is important to note that the Regional Adaption Spectra-Phenology Integration (RASP)
soybean mapping strategy involves several key steps, including potential area identification, feature
selection, unsupervised learning, and cluster assignment. Finally, we conducted multi-comparisons
between our soybean products with others, including census data, ground samples, and existing datasets,
to evaluate the accuracy of our data product.

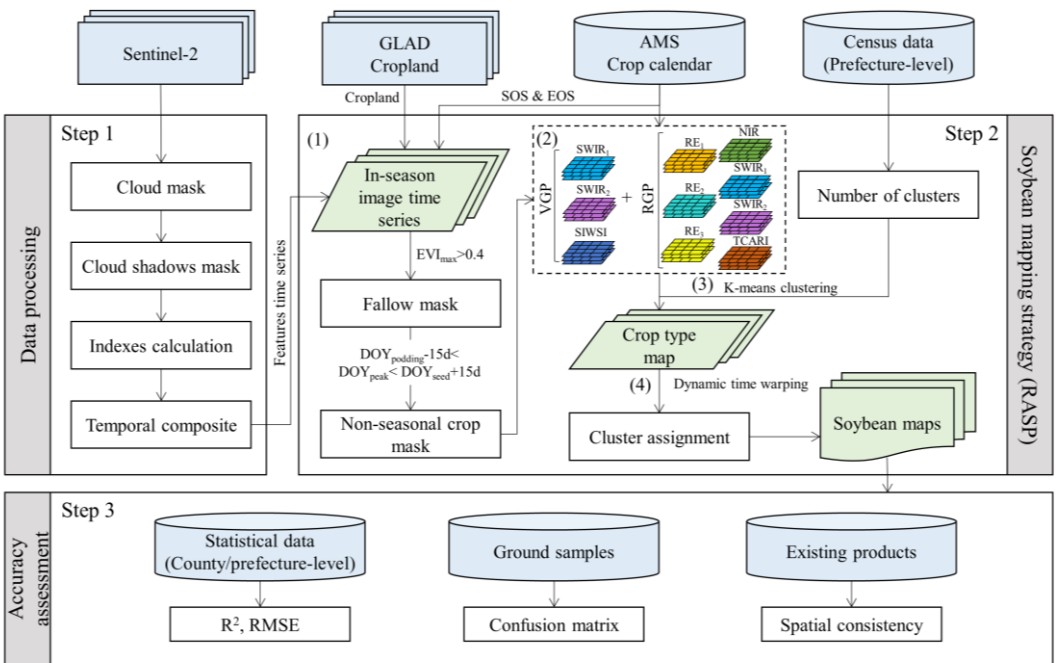

Figure 2. The Regional Adaption Spectra-Phenology Integration methodology for retrieving soybean planting area. AMS, agricultural meteorological station; $DOY_{podding}$, the podding date recorded by the nearest AMS; EVI: Enhanced Vegetation Index; $DOY_{peak}$, the date when EVI reached peak; $DOY_{seed}$, the full-seed date recorded by the nearest AMS; SOS, start of growing season; EOS, end of growing season; $SWIR_1$, Short Wave Infrared band 1; $SWIR_2$, Short Wave Infrared band 2; SIWSI, shortwave Infrared Water Stress Index; $RE_1$, Red Edge band 1; $RE_2$, Red Edge band 2; $RE_3$, Red Edge band 3; NIR, Near-infrared band; TCARI, Transformed Chlorophyll Absorption in Reflectance Index; VGP: vegetative growing period; RGP: reproductive growing season.

**2.3.1 Data processing**

We employed the simple cloud score algorithm (Oreopoulos et al., 2011), QA60 band, cirrus band, and cloud probability dataset to identify cloud masks. The following isolated cloud masks are created: (1) Cloud and cirrus identified by QA60 band; (2) Cirrus identified by cirrus band in Level-1C products; (3) Pixels with cloud score less than 0.9; and (4) Pixels with cloud probability more than 70. Each algorithm has its own strengths and limitations. For example, QA60 band removes a large number of thin cirrus clouds while ignoring small clouds with thicker resolution, and the fixed threshold values of cloud score and cloud probability may introduce uncertainties. Therefore, we masked the pixels identified as clouds by at least two methods to achieve better cloud removal effects. Then, we used Temporal Dark Outlier

Mask (TDOM) method to eliminate cloud shadows (Housman et al., 2018). We calculated the SIWSI
and TCARI indices based on the Sentinel-2 image set processed above (see 2.3.2(2)). To fill the data gaps
caused by cloud removal and smooth anomalies, Sentinel-2 time series was reconstructed by moving
median composite method, resulting in a 10-day interval composite time series. We set the half-window
size for the moving median methods to 10 days considering the 5-day revisit cycle of Sentinel-2 and
computational efficiency. In areas with notably limited clear observations, a gap-filling method was
conducted on the composite time series. This method involves substituting any given observation with
the median value from three neighboring observations (i.e., previous, current, and subsequent
observations) to maximize the continuity and completeness of time series.

**2.3.2 Regional Adaptation Spectra-Phenology Integration (RASP) soybean mapping strategy**

(1) Potential area identification
To minimize the impact from non-croplands, we firstly determine the potential cropping areas by
masking GLAD cropland layer over study area. Sentinel-2 images within growing season were extracted
by taking the sowing date and harvesting date recorded at the nearest agricultural meteorological station
(AMS) as the starting and ending dates of the growing season, respectively. Based on the cropland
extracted, we filtered out the pixels exhibiting an Enhanced Vegetation Index (EVI) maximum value
during the growing season less than 0.4 to remove fallow land according to the analysis of ground
samples (Fig. S1) and previous studies, which found that almost all crops had maximum EVI values
above 0.4 (Li et al., 2014; Zhang et al., 2017; Han et al., 2022). EVI is a vegetation index with high
sensitivity in biomass:

$$EVI = G \times \frac{\rho_{NIR} - \rho_{Red}}{\rho_{NIR} + C_1 \times \rho_{Red} - C_2 \times \rho_{Blue} + L} \tag{1}$$

Where $\rho_{NIR}$, $\rho_{Red}$, and $\rho_{Blue}$ represented the reflectance of the Near-infrared (835.1nm (S2A) / 833nm
(S2B)), Red (664.5nm (S2A) / 665nm (S2B)), Blue (496.6nm (S2A) / 492.1nm (S2B)), respectively.
The greenest period of soybean typically occurs between the podding date and the full-seed date, with a
difference of more than a month from the peak date of non-seasonal crops, such as wheat (Fig. 4a). We
obtained the phenological observations recorded by the nearest AMS as reference and set the restricted
time window from 15 days before the podding date (DOY$_{podding}$) to 15 days after the full-seed date
(DOY$_{seed}$) (Fig. 3). We generated the potential area by eliminating pixels whose EVI maximum occurs
outside the given time window because the phenological difference of soybeans in adjacent areas
generally does not exceed one month. Moreover, the impacts of cloud-covered pixels appearing in the
proposed period is minimized since we have reconstructed the original EVI time series.

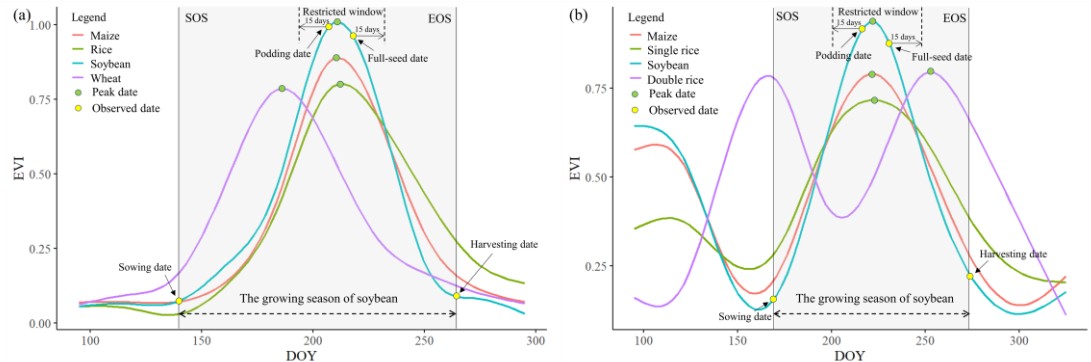


**Figure 3. Schematic diagram of seasonal crop identification for (a) single - and (b) double - cropping systems.**
(2) Feature selection
By exploring the spectral characteristics of crop field samples, we identified reflectance bands and
vegetation indices that are significantly associated with soybeans but different from other crops. We
selected six bands and two spectral indices for crop mapping, including Near-infrared (NIR) band, Red
edge band 1 (RE1), Red edge band 2 (RE2), Red edge band 3 (RE3), Short Wave Infrared band 1
(SWIR1), Short Wave Infrared band 2 (SWIR2), Shortwave Infrared Water Stress Index (SIWSI),
Transformed Chlorophyll Absorption in Reflectance Index (TCARI). SIWSI is an indicator of canopy
water content that reflects soil moisture variations and canopy water stress better than Normalized
Difference Vegetation Index (NDVI) (Fensholt and Sandholt, 2003; Olsen et al., 2015). TCARI is an
indicator which is sensitive to chlorophyll concentration (Sobejano-Paz et al., 2020). The two spectral
indices were calculated as follows:

$$SIWSI = \frac{\rho_{SWIR1} - \rho_{NIR}}{\rho_{SWIR1} + \rho_{NIR}} \tag{2}$$

$$TCARI = 3 \times ((\rho_{VRE1} - \rho_{Red}) - 0.2 \times (\rho_{VRE1} - \rho_{Green}) \times \rho_{VRE1}/\rho_{Red}) \tag{3}$$

Where $\rho_{SWIR1}$,$\rho_{NIR}$,$\rho_{VRE1}$,$\rho_{Red}$ and $\rho_{Green}$ represented the reflectance of the Short Wave Infrared
band1 (SWIR1, 1613.7nm (S2A) / 1610.4nm (S2B)), Near-infrared (835.1nm (S2A) / 833nm (S2B)),
Red Edge1 (VRE1, 703.9nm (S2A) / 703.8nm (S2B)), Red (664.5nm (S2A) / 665nm (S2B)), Green
(560nm (S2A) / 559nm (S2B)), respectively.
During early growing season of soybean (~DOY 120-190 in Zone I), the flooding signal of rice was
obvious due to the transplanting period. This resulted in a significantly lower SWIR reflectance and
SIWSI index for rice compared to those of soybean (Fig. 4f-h). SWIR bands and SIWSI index during the
vegetative growing period (VGP) of soybean can effectively distinguish dryland crops (such as soybean,
maize) from paddy crops (such as rice).
Soybean has a lower water content during the middle and later growing season (~DOY 190-220 in
Zone I) than maize, resulting in higher reflectivity in SWIR bands (Fig. 4b, 4f, 4g) (Chen et al., 2005). It
has been demonstrated that SWIR and red-edge bands can effectively differentiate soybean and maize
(Fig. 4c-g) (Zhong et al., 2016; You and Dong, 2020; Liu et al., 2018b). Additionally, the chlorophyll
content of soybean in the middle and late growth period was lower than that of maize, leading to
significantly higher TCARI values. Meanwhile, the timing of TCARI reaching saturation significantly
differs among soybean, rice, and wheat (Fig. 4i). All these spectral-phenological characteristics are also
applicable to soybeans planted in other sub-zones (Fig. S2-S4). Based on these findings, we selected NIR,
red-edge bands, short-wave infrared bands, and TCARI index during soybean reproductive growing
season (RGP) as key features.

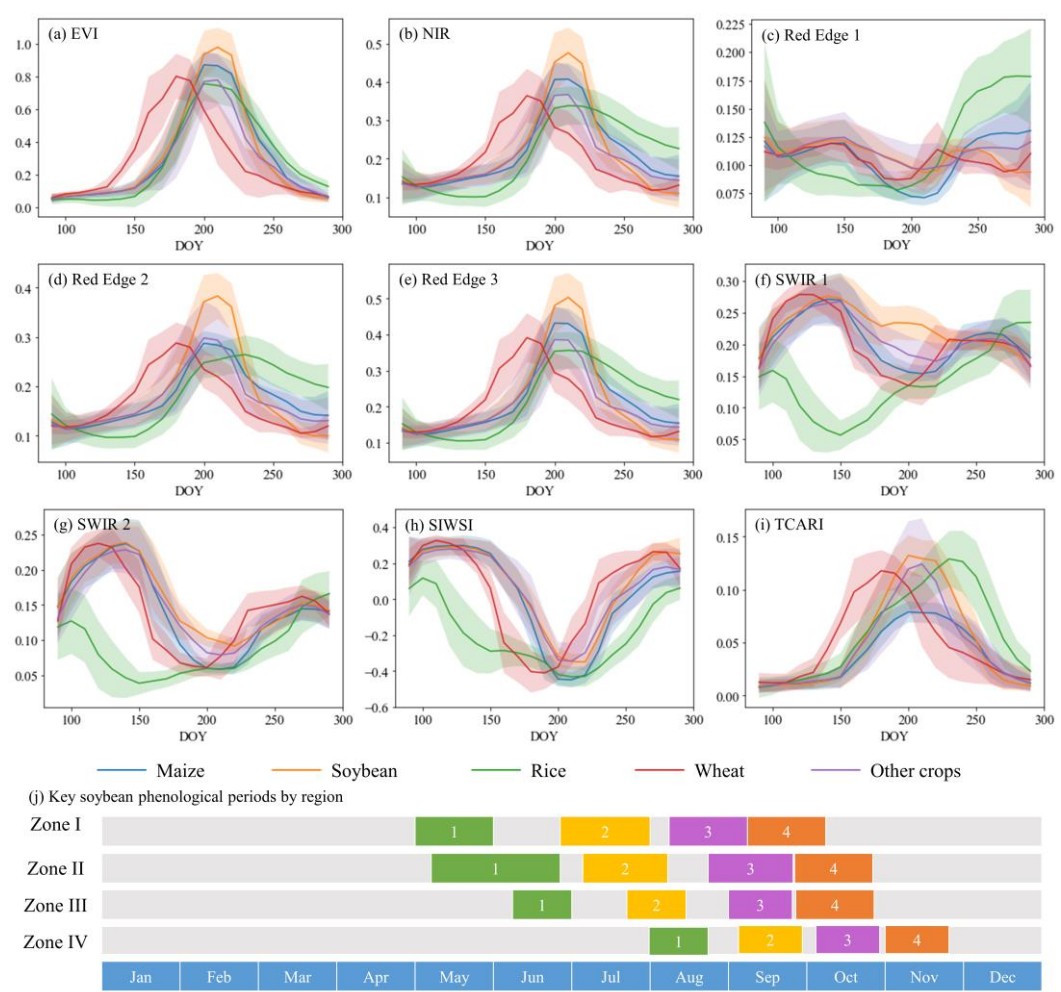


**Figure 4. Temporal profiles of (a-i) for major crops in Northeast China and (j) key soybean phenological periods by region based on ground samples. Lines depict the mean values of different crops and shaded areas depict error bars with one positive/negative standard deviation. The number at the bottom represents the key phenological periods of soybean: 1 – Sowing, 2 – Flowering, 3 – Seed fulling, 4 – Maturity.**

(3) Unsupervised learning

We utilized K-means algorithm to classify potential area data by using the wekaKMeans Clusterer provided by Google Earth Engine (GEE). The $m$ samples are divided into $k$ clusters by alternately assigning the samples to the nearest cluster centroid measured by Euclidean distance or the Manhattan distance and updating the cluster centroid to the mean of the samples assigned to the cluster. This approach had been widely used in land-cover classification and crop mapping (Xiong et al., 2017; Wang et al., 2019). We used the detailed phenological records at AMSs to identify soybean growth periods and selected the spectra and vegetation indices within specific growth periods (VGP, RGP) as input features. The classifier was trained individually on each prefecture based on the number of clusters $k$ input. The cluster number $k$ is defined as the number of "major crops" that constituting 95% of the total area for seasonal crops (including rice, maize, soybean, cotton, peanuts, sesame, sweet potato, and sorghum) according to prefecture-level statistics, and plus one for "other crops".

(4) Cluster assignment

To identify the most likely cluster that represents soybean, we randomly selected 100 points per cluster and extracted feature series. We then used dynamic time warping (DTW) method to measure the similarity between each cluster's eight features involved in classification and the soybean standard curves. We averaged the data of 30% samples in each sub-zone to establish the standard curves, reducing the impact of regional phenological variations. The time coverage of Zone I-IV was set to April-September, May-October, June-October, and August-November, respectively, which are corresponding with the soybean growing season. The cluster with the minimal average of 8 DTW values was identified as the soybean cluster. DTW is a flexible algorithm that allows for deviations in time between two sequences, and it calculates the minimum distance between them by finding misalignment matches between elements. This approach is widely used in land cover and crop identification due to its ability to handle time distortions associated with seasonal changes (Guan et al., 2016; Dong et al., 2020).

### 2.3.3 Accuracy assessment

To assess the accuracy of the soybean maps we generated, we validated and compared the results using 1) county- and prefecture-level census data, 2) ground samples, and 3) existing products. Since the county-level statistics after 2019 were not fully collected, we used the county-level statistics for 2017-2018 and the prefecture-level statistics for 2019-2020 to calculate the $R^2$ and RMSE of the mapped area with the following equations:

$$R^2 = 1 - \frac{\sum_{i=1}^{n}(s_i - y_i)^2}{\sum_{i=1}^{n}(s_i - \bar{s})^2} \tag{4}$$

$$RMSE = \sqrt{\frac{\sum_{i=1}^{n}(s_i - y_i)^2}{n}} \tag{5}$$

where $s_i$ and $y_i$ are the statistical and mapped soybean area for county (prefecture) $i$, $\bar{s}$ is the average statistical area, and $n$ represents the total number of counties (prefectures). We calculated the local crop mapping area based on the Universal Transverse Mercator (UTM) projection corresponding to the location of the province.

We also used ground samples during 2017-2019 to verify the authenticity of the soybean maps. Confusion matrices were calculated as follows:

$$PA = \frac{N_i}{R_i} \tag{6}$$

$$UA = \frac{N_i}{C_i} \tag{7}$$

$$OA = \frac{N_c}{A} \tag{8}$$

$$F1 = 2 \times \frac{UA \times PA}{UA + PA} \tag{9}$$

where $N_i$ is the number of correctly identified validation samples of class $i$, $R_i$ is the number of ground validation samples of class $i$, $C_i$ is the number of validation samples classified as class $i$, $C_i$ is the number of validation samples classified as class $i$, $N_c$ is the total number of correctly identified validation samples, $A$ is the total number validation samples. $PA$, $UA$, and $OA$ represent producer's accuracy, user's accuracy, and overall accuracy, respectively.

To ensure that the products are accurate not only in quantity but also in space, we further compared the ChinaSoyArea10m with existing products in detail space.

**3 Results**

**3.1 Accuracy assessment**

We utilized the available census data from 2017-2020 (at county-level in 2017-2018 and prefecture-level in 2019-2020) to verify the accuracy of the soybean maps across the entire studied area. Annual ChinaSoyArea10m is consistent well with the census data ($R^2 > 0.8$), with an $R^2$ value of 0.84, 0.85, 0.82, and 0.86 for 2017, 2018, 2019, and 2020, respectively (Fig. 5). These results demonstrate that our RASP method is inter-annual robustness and can accurately capture annual dynamics of soybean planting areas. The scattered points are generally distributed around 1:1 line, without large overestimations or underestimations. However, the areas are overestimated for counties with planting area < 20 kha, or prefectures with planting area <100 kha (Fig. 5). This uncertainty, particularly overestimation, could be caused by the low proportion of soybean cultivation. If maize or other same-season crops are planted in a much higher proportion than soybeans there, distinctly recognizing soybeans (as a less prevalent crop) as a separate category will be a big challenge for classifiers, consequently resulting in misclassified clusters including maize or other crops.

The mapping accuracy in Zone I closely matched county-level statistics, showing high consistency ($R^2$=0.86). Zones II-IV also demonstrated reasonable agreement ($R^2$=0.50~0.69), despite relatively lower accuracy due to the scarcer planted areas (Fig. S5). No significant trend deviation from statistics was indicated for the mapping area in Zone I, with slight overestimates for Zone II and III, and underestimates for Zone IV (Fig. S5). These accuracy variations are acceptable, given the challenges in accurately identifying soybeans in regions where they are planted less prevalently. Specifically, maize is more dominant than soybeans in Zone II, while Zone III is characterized by diverse crops and complex planting patterns. Underestimation in Zone IV is possibly due to fewer clear observations in the southwest. Nevertheless, the overall accuracy across the zones is acceptable.

ChinaSoyArea10m is consistent well with census data compared to the existing product (CDL) (You et al., 2021), using both the county level in 2018 and prefecture level in 2019 (Fig. 6). CDL's results are consistent with census data at the prefecture scale, with more overestimations at the county level (Fig. 6), implying the comparison at finer scale would reveal more details. ChinaSoyArea10m is consistent

with statistics at the both levels ($R^2$ ~0.85), with $R^2$ increases 0.31 compared with CDL in county level
(Fig. 6a).

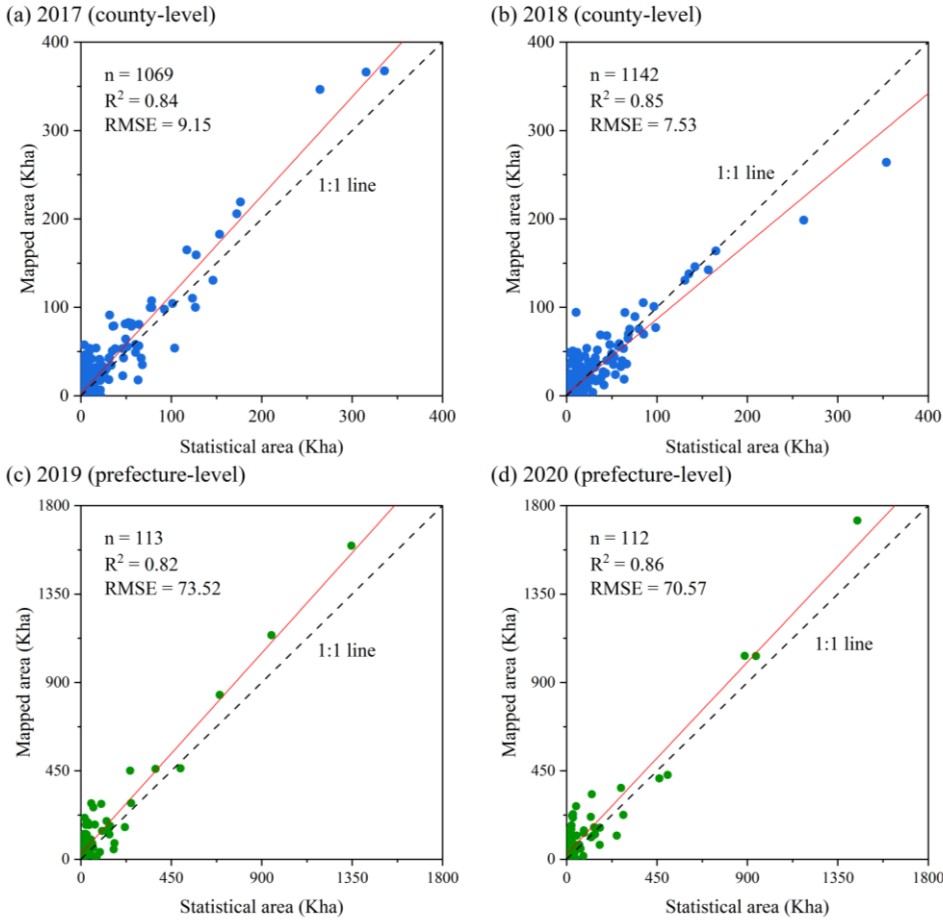


**Figure 5. Comparison of soybean areas with statistics in (a) 2017 at county-level, (b) 2018 at county-level, (c)**
**2019 at prefecture-level, (d) 2020 at prefecture-level.**

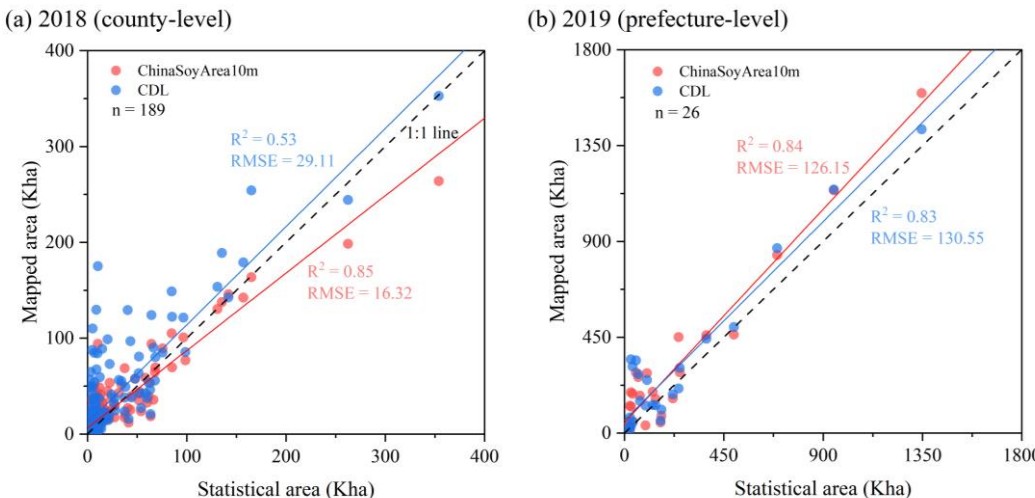


**Figure 6. Comparison of soybean areas of ChinaSoyArea10m and CDL with statistics in (a) 2018 at county-**
**level, (b) 2019 at prefecture-level.**
Furthermore, we used ground samples in 2017-2019 to validate the reliability of the soybean maps.
Since the soybean planting area maps are 0-1 binary images, we categorized the ground samples into
soybean and non-soybean (maize, rice, wheat, and other crops). The verification results based on ground
samples indicated that the overall accuracy of soybean maps during 2017-2019 was in the range of 77.08%
to 86.77%. The F1 scores of soybeans increased from 2017 to 2019 (0.69, 0.75 and 0.84, respectively)
(Table 2). The variance in accuracy among years could be attributed to the quality of Sentinel-2 images,
which had been indicated in previous studies (Liu et al., 2020; Han et al., 2021). The overall accuracy
for each sub-zone in 2019 varied from 83.58% to 90.67% (Table S1). Specifically, Zone I demonstrated
the highest producer's accuracy for soybean at 88.31%, aligning with its high consistency with statistics.
Zone III achieved the highest overall accuracy at 90.67%, attributed to its superior user's accuracy for
soybean, indicating fewer misclassifications, and effective differentiation from non-soybean crops (Table
S1). The producer's accuracy in Zone IV was relatively lower at 63.89%, possibly due to the limited
samples, high heterogeneity, and fewer clear observations (Table S1).

381                  **Table 2. Confusion matrix of the soybean maps during 2017-2019.**

| | Reference | Map Soybean | Non-Soybean | Producer's Accuracy | User's Accuracy | F1 Score | Overall Accuracy |
|---|---|---|---|---|---|---|---|
| 2017 | Soybean | 679 | 352 | 65.86% | 72.47% | 0.69 | 77.08% |
| | Non-Soybean | 258 | 1372 | 84.17% | 79.58% | 0.82 | |
| 2018 | Soybean | 799 | 246 | 76.46% | 74.19% | 0.75 | 85.16% |
| | Non-Soybean | 278 | 2208 | 88.82% | 89.98% | 0.89 | |
| 2019* | Soybean | 1279 | 235 | 84.48% | 83.32% | 0.84 | 86.77% |
| | Non-Soybean | 256 | 1940 | 88.34% | 89.20% | 0.89 | |

* Including ground samples and nationwide reference points based on existing datasets.
**3.2 Spatial distributions of soybean planting areas**
Based on the soybean maps, we further analyzed the spatial patterns of soybean distribution in China
during 2017-2021. There were small changes in the spatial distribution of soybean in China in recent
years (Fig. 7-8). Several hot spots were obviously observed in Heilongjiang Province, eastern Inner
Mongolia, and northern Anhui, especially for eastern Inner Mongolia and western Heilongjiang,
extensively and densely distributed by soybean fields (Fig. 8b-c). In Region II, soybean was planted at a
larger scale, mainly concentrated in northern Anhui (Fig. 8d), and extensively distributed in Henan and
Shandong (Fig. 8e). Soybeans in other provinces of Region II, III, and IV were scattered distribution,
especially in the southwestern mountainous region (Fig. 8f-h).

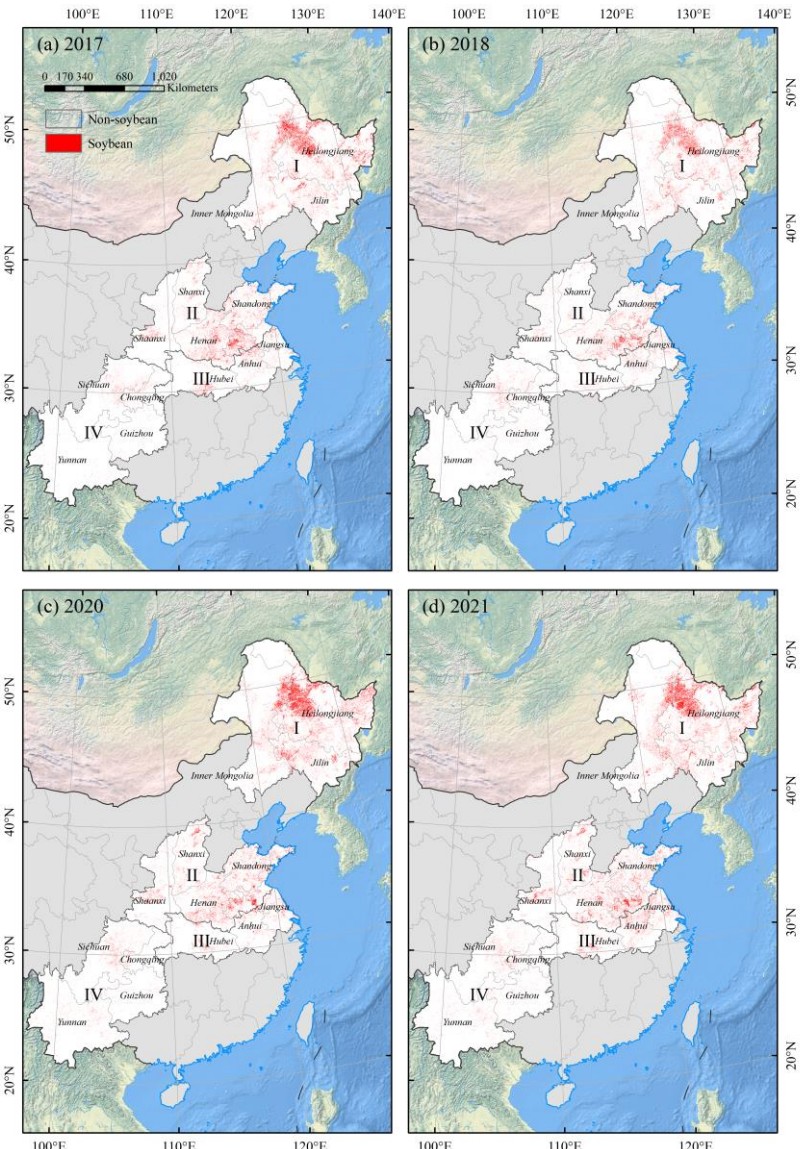


**Figure 7. Spatial distribution of soybean areas at 10 m resolution across China in (a) 2017, (b) 2018, (c) 2020**
**and (d) 2021.**

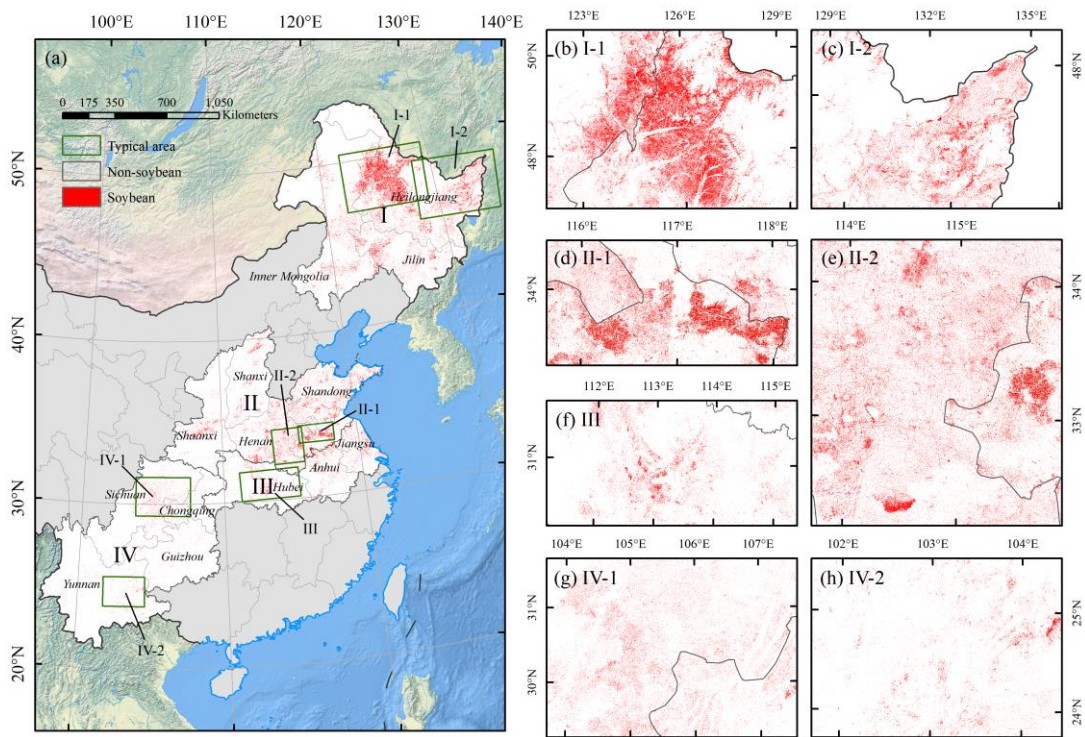


**Figure 8. Spatial distribution of soybean areas at 10 m resolution across China (a) and zoom-in maps of each region (b-h) in 2019.**

To further compare soybean maps in detail, we compared ChinaSoyArea10m with GLAD maize-soybean map and CDL data products in space. The GLAD product is a 10-m resolution maize-soybean map of China in 2019, and their $R^2$ values with provincial and prefecture statistics were reported by 0.93 and 0.94 (Li et al., 2023). Arable land near waterbodies is often misclassified as soybean plots by CDL, which has not occurred by GLAD and ChinaSoyArea10m, implying other crop types are possibly misclassified as soybeans by CDL (Fig. 9 a1-d1). As for the second case (Fig. 9 a2), our extraction results are similar to those of GLAD, while small plots failed to be identified by CDL (Fig. 9 a2-d2). In areas where banded soybeans are planted less concentrated, CDL tended to overestimate the soybean area (Fig. 9 a3-d3), further substantiating the above limitations (Fig. 6). Conversely, our mapping results behaved similarly as GLAD did (Fig. 9 a3-d3). The overall accuracy of GLAD map based on pure samples reaches 95.4% (Li et al., 2023), so GLAD can be regarded as a reliable reference. From the three cases, therefore, ChinaSoyArea10m has behaved more similarly with GLAD than CDL does, indicated by less underestimation, less overestimation, and higher accuracy in details.

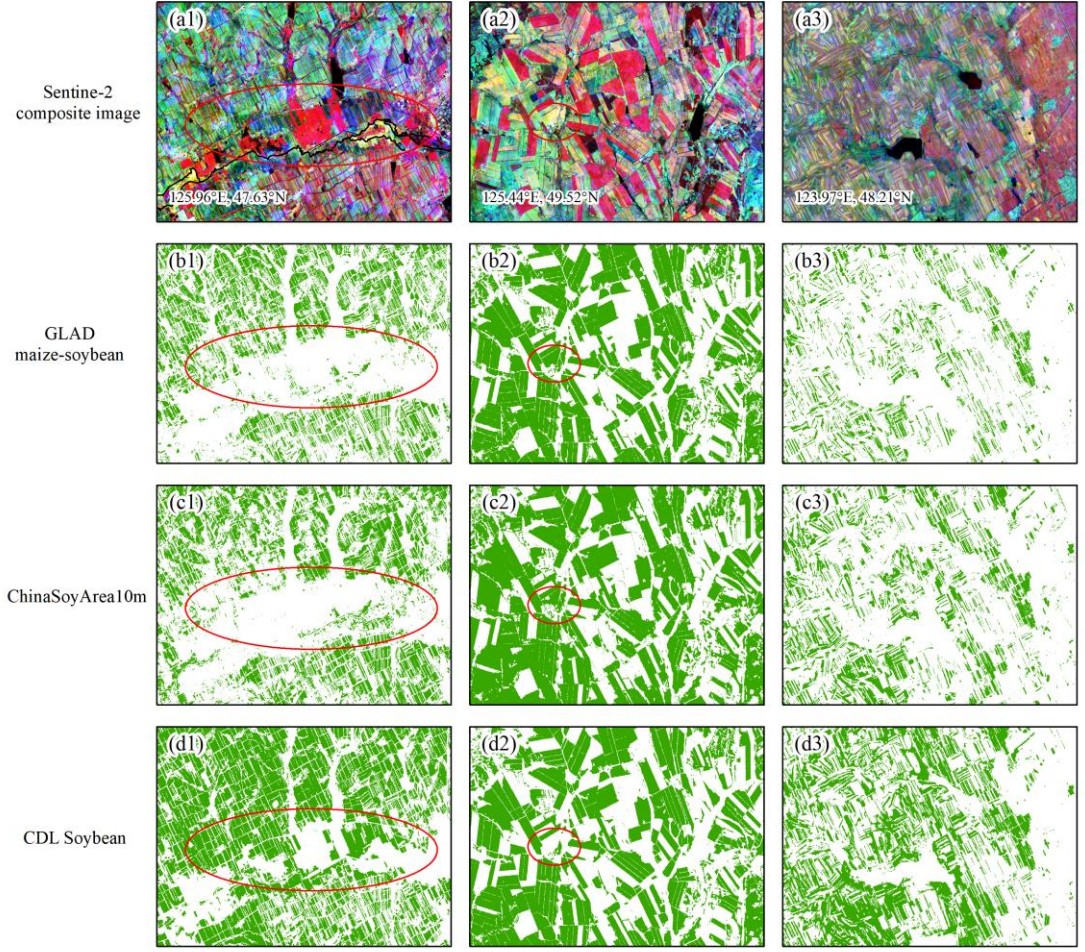

**Figure 9. Visual comparison of our soybean maps and existing products in typical regions in 2019: (a1-a3) RGB composite images comprise NIR (Band 8), SWIR 2 (Band 12), and SWIR 1 (Band 11) bands from Sentinel-2 median composite images during the peak growth period of soybean; (b1-b3) soybean layer extracted from GLAD maize-soybean map; (c1-c3) ChinaSoyArea10m map; (d1-d3) soybean layer extracted from CDL.**

## 4 Discussion

### 4.1 Our advantages and potential applicability

We proposed a new framework (RASP) to identify annual dynamic of soybean planting areas over larger regions and produced the longer-term series of soybean maps (ChinaSoyArea10m) across mainly planting areas in China from 2017 to 2021 at the first time. The accuracy of ChinaSoyArea10m is acceptable ($R^2 \sim 0.85$) at both county- and prefecture-level, with relatively less

$R^2$ than GLAD ($R^2 = 0.93$ at prefecture-level), but higher than CDL ($R^2 = 0.53$ at county-level). Compared with existing products, ChinaSoyArea10m accurately depict the soybean with more spatial and temporal details as well.

The methodology developed for identifying soybean planting areas indicate several notable strengths that make it an attractive option for wide application. Firstly, it operates independently, without extensive ground samples required. The conventional supervised approaches like random forest (RF) and long short-term memory (LSTM) depend on quantities of observations, with much money, time, and labor consumed. In this context, both transferable learning model and our RASP methods (combing unsupervised learning with statistics) indeed provide huge potential for crop mapping. However, transferable models are suitable for areas or years with similar cropping patterns. In areas with diverse and complex cropping patterns, it is a challenge to apply the supervised model trained in limited areas or limited years into others (Wang et al., 2019; Ma et al., 2020). In contrast, our strategy leverages a specific, pre-existing set of samples to stably differentiate soybean characteristics from other crops, which can accurately map annual dynamics without updated requirement in annual samples. Consequently, this method significantly weakens limitations in crop classification during years without specific samples, enabling crop mapping consistently and continually.

Another key advantage of our spectra-phenology integration approach is its quick applicability over larger areas, coupled with excellent spatial scalability. It can self-adopt to different environments by considering phenology information. Compared to methods that rely on composite indicators and specific thresholds, our approach simplifies the requirements for inputs and experienced judgements. The only inputs required are the phenological information of soybeans and the number of other primary crops during the same growing season in the targeted area. This allows to classify crop swiftly and efficiently without additional inputs for background knowledge or setting complex thresholds. The input of phenological information in each prefecture enhanced the zonal adaptive assessment of soybean growth status across various areas, thereby facilitating crop classification. This innovative approach ensures its applicability into other soybean-producing areas, showcasing its potential for broader implementation.

## 4.2 The uncertainty from image quality

The method we proposed (RASP) is strongly dependent on remote sensing images and subregional unsupervised classification by considering the bands and vegetation indices, which are all sensitive to the unique characteristics of soybeans. Therefore, the accuracy of soybean maps inevitably is associated with the quality of remote sensing images. By using ground samples to validate the mapping results, we found that the accuracy of 2017 is lower than that of 2018 and 2019, with an overall accuracy is less than 80% (Table 2).

We extracted cloud-free images in different regions during the soybean growing season and calculated the monthly average number of clear observations. In general, the monthly averages of clear observations in Northeast region and Huang-Huai-Hai region (Zone I and Zone II) are relatively higher than the southern zones (Zone III and IV) (Fig. 10a2-e2). In areas with quite lower clear observations, despite a gap-filling method was conducted to generate complete 10-day composite time series, higher uncertainty is inevitable. The gap-filling time series might contain duplicate values, which cannot accurately reflect the crop growth process in reality. Obviously, the total number of images available in 2017 over the study areas was significantly fewer than those of other years, because the second satellite Sentinel-2B only commenced operations and started providing data after March of 2017 (Fig.10a1-e1). Removing the cloudy pixels has left ever fewer clear images available (upper vs. down layer in Fig.10). During the growing season, the average number of clear observations per month was 0-2 in partial regions, lower than the requirements of 10-day time series composite we mentioned in 2.3.1. This might explain the lower user's accuracy of soybean in Zone IV compared to other sub-zones (Table S1) and low overall accuracy based on sample verification in 2017 (Table 2).

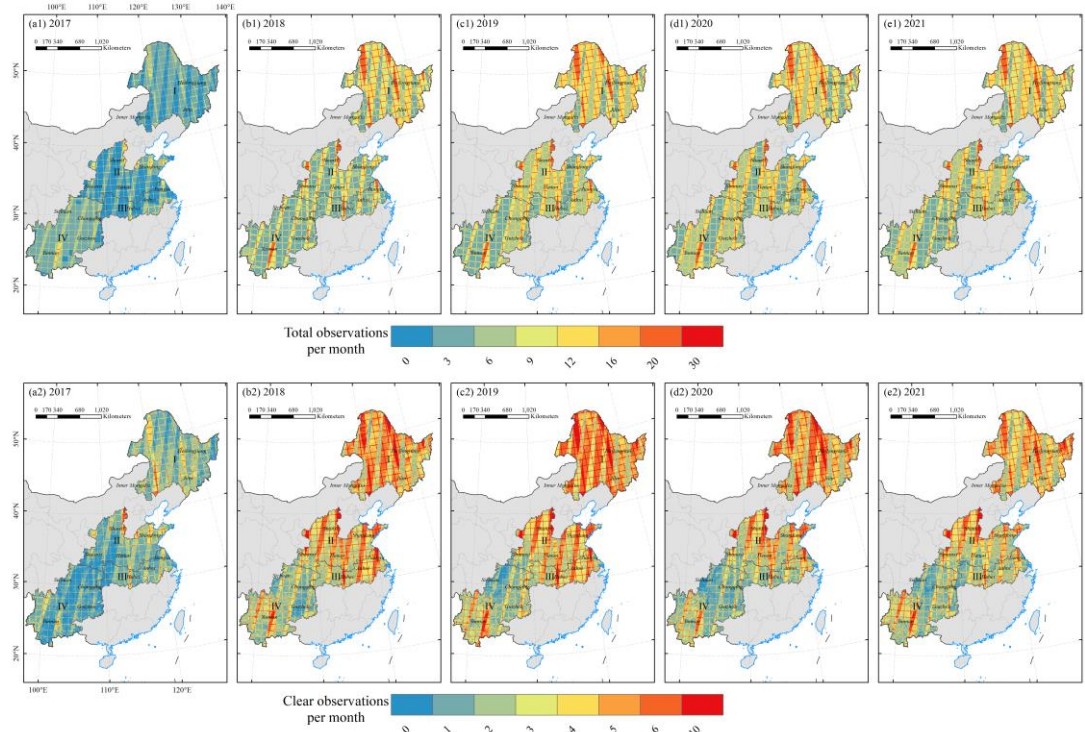


**Figure 10. Total (a1-e1) and clear (a2-e2) observations per month during soybean growing season.**

**4.3 Limitations in small-scale planting areas**
Validation based on statistics shows that ChinaSoyArea10m reached a high consistency ($R^2 \sim 0.85$) across
China. However, in areas with soybean sparsely planted, the consistency is lower than that in densely
planted areas, with more overestimations observed in the sparse areas. Such overestimations are caused
by the limitations of unsupervised classification algorithm. K-means is difficult to accurately capture
small plots of crops in a complex cropping system, although it can make up for the shortage of crop
mapping in some areas with limited training samples (Kwak and Park, 2022). Studies have proved that
the classifier performs inferiorly where dominant crop phenotypes are similar, and crop diversity is higher
(Wang et al., 2019; Konduri et al., 2020). Therefore, the classifier is challenged in areas where soybean
is not the dominant type due to the small plot size and spectral overlap between different crops (Chabalala
et al., 2022). In southern China, cropland plots are typically small (<0.04 ha in most regions) and the
crop diversity is high. The growth periods of soybean, peanut, potato, and maize are similar, dominantly
indicated by a mixed planting pattern, which has contributed to the low accuracy of non-main soybean
producing areas in southern China (Liu et al., 2020). Additionally, soybeans are intercropped with maize
or other crops in some areas, where the strip width is less one meter (Yang et al., 2014; Du et al., 2018).
This planting pattern will introduce the mixed pixels problem as well under the background of 10 m
resolution crop mapping.
The lower accuracy in soybean area sparely planted could be explained by the characteristics of K-
means algorithm. K-means algorithm is developed to minimize the distance between each point within a
cluster and the cluster's centroid. When the sample size in a particular category substantially exceeds
those of others, the algorithm might preferentially optimize the cohesion of the larger category, and would
neglect the accurate clustering for smaller categories (Tan et al., 2016). The effectiveness of K-means
classification is highly dependent on the selection of initial clustering centers. In scenarios of unbalanced
categories, initial centers randomly selected might inadequately represent the minor categories, resulting
in inaccurate results (Tan et al., 2016). Additionally, K-means assumes that each cluster is spherical;
therefore, it does not perform well when clusters are non-spherical and uneven in size and density. Hence,
in areas with unbalanced crop categories, the algorithm faces challenge to assign each crop to a
corresponding cluster precisely (Tan et al., 2016; Wang et al., 2019).
Our regional adaptive large-area crop mapping method in future will further be improved by the
follows: (1) Classification on a finer scale by specifying a more precise number of target clusters can
reduce spatial heterogeneity and emphasize the relative importance of non-dominant categories, and
increase classification accuracy consequently (Li and Yang, 2017). (2) Optimizing data preprocessing
methods. Outliers can interrupt classification because the unsupervised methods is highly sensitive to
anomalies (Raykov et al., 2016; Wang et al., 2019). Therefore, eliminating outliers can further improve
the classification validity. In addition, since K-means weights all dimensions equally, minimizing the
features' correlation and reducing irrelevant variables are also important means to enhance the
classification effect (Hastie et al., 2009). (3) Improving algorithm performance. A variety of algorithms
have been proposed to address the inherent defects of K-means (Ahmed et al., 2020), such as by
optimizing the initial clustering center (e.g., K-means++), weighting classes (e.g., Weighted k-means),
and non-spherical clustering assumptions (e.g., DBSCAN, Spectral Clustering) (Ester et al., 1996; Bach
and Jordan, 2003; Kerdprasop et al., 2005; Arthur and Vassilvitskii, 2007). The improved algorithms will
address the issues on complex and highly diverse crop classification in some degrees (Li et al., 2022;
Rivera et al., 2022). (4) Better post-processing of data. Misclassification of field ridges and image
speckles is inevitable during mapping crops over large areas. With the progress of computing power,
auxiliary data and image processing algorithms can further eliminate these issues (Liu et al., 2018a; Li
and Qu, 2019; Hamano et al., 2023). We are sure that integrating cloud computing platforms with
advanced algorithms will provide substantial potential for accurate crop identification covering larger
areas in future.

**5 Data availability**


The soybean planting area product for China during 2017-2021 (ChinaSoyArea10m) is available at
https://zenodo.org/doi/10.5281/zenodo.10071426 (Mei et al., 2023). We encourage users to
independently verify data products for special study areas before using them.

**6 Conclusions**


In this study, a Reginal Adaption Spectra-Phenology Integration (RASP) method over large-scale was
developed and utilized to generate soybean planting area maps for major producing regions in China
from 2017 to 2021. By utilizing Sentinel-2 images, spectral features and vegetation indices that best
distinguish soybeans were extracted and input into an unsupervised classifier in each prefecture. The
DTW method was then employed to identify the soybean distribution. RASP does not rely on many
ground samples and considers the soybean phenology in various planting areas, suggesting a potential
way for long-term crop mapping over larger regions. Verification results demonstrated a high consistency
between the mapping results and census data at county or prefecture level (all > 0.82), with overall
accuracies of field samples reaching 77.08%~86.77%. These findings confirm the reliability of
ChinaSoyArea10m. Our data products fill the gap in regional long-term soybean maps in China, and
provide important information for sustainable soybean production and management, agricultural system
modeling, and optimization.

**Author contributions.**


ZZ and FT conceive this study. QM, JH, and JD collected datasets. QM implemented the research and
wrote the original draft of the paper. All authors discussed the results and revised the manuscript.
**Competing interests.**
The contact author has declared that neither they nor their co-authors have any competing interests.
**Financial support.**
This research was funded by the National Key Research and Development Program of China
(2020YFA0608201) and National Natural Science Foundation of China (42061144003, 41977405).

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
