# Peer review of "ChinaSoyArea10m: a dataset of soybean planting areas"

_Earth System Science Data, 2023_

## Referee Comment (RC4)

**Comments**

Title: ChinaSoyArea10m: a dataset of soybean planting areas with a spatial resolution of 10 m across China from 2017 to 2021
Author(s): Qinghang Mei et al.
MS No.: essd-2023-467
MS type: Data description paper

Mei et al's work mapped the soybean planting areas across China with a high spatial resolution of 10 meters, spanning from 2017 to 2021, provided important information for sustainable soybean production and management, as well as agricultural system modeling and optimization. In this work, authors summarized five methods of mapping crops by remote sensing. The advantages and uncertainties of each method were compared, and a highly effective for accurately mapping crops over a larger region method named combining unsupervised classification and post-classification methods applied in this paper. They accomplished this by Sentinel-2 remote sensing images from the GEE platform with cropland layer and detailed phenology observations. They validated the results with the census data at both county- and prefecture-level, and with the two existing datasets (CDL and GLAD maize-soybean map).

Overall, I find this work to be valuable. However, I have some concerns regarding the robustness from the sparse number of AMSs in SW Zonal IV and uncertainty in quality of satellite imagery. I hope the authors will consider these points and provide further clarification in their responses and/or revisions. Please find my major comments and minor for clarification below.

**Major comments:**
1. The text mentions the need for 10-day time series composite images per month, but in certain areas, the average monthly count of clear observations is insufficient to meet this requirement. Can the existing time series composite methods be optimized to accommodate the inadequacy of observational data?
2. The observations per month of satellite imagery in SW Zonal IV are less, and the AMSs in this zonal also only have two sites. Whether it is possible to increase the observational data or phenological data from remote sensing to test the robust.
3. To determine the potential cropping areas, authors filtered the pixels exhibiting an EVI maximum value during the growing season greater than 0.4 to remove fallow land. For spatial variation across four zonal, the constant threshold would bring some uncertainty. I expect to see more evidence for selecting 0.4 or a sensitivity analysis of threshold can also be implemented.

**Minor comments:**
Line 58: "same areas" means the north China?
Line 180, Figure2: The label on the left in Figure2 (i.e. 'Data processing' and 'Accuracy assessment') are set to rotate 180° to match reading habits.
Line 180, Figure2: In step2, part (2) of the dashed box is confusing. What the color

represents? If I understand correctly, they represent different layers of indexes. It is recommended to put the abbreviation to the right of the color layers.

---

## Community Comment (CC1)

**Response to reviewer #1:**

General comment: The article proposed an unsupervised method for identifying soybean crops within the defined croplands across China. The topic is interesting, and also important for sustainable agricultural development due to its large spatial and long-term coverage. The data is well collected and processed, and the results are properly presented. I would suggest some minor revisions as listed below.

Thank you for your positive and constructive comments, which surely encourage us to further enhance our research quality. We carefully revised our manuscript and provided a point-by-point response below. Moreover, we have positively addressed all points in the revised edition, which will be updated after responding all referees' comments.

Comment 1: L43: Not sure what are the "shortcomings of domestic supply"?

Sincerely apologize for the ambiguousness here. We have changed "shortcomings" into "shortages" throughout the manuscript.

We have further elucidated the issue of soybean supply in China in our revised manuscript. The shortages of soybean supply in China are evident in its growing dependence on imports and the decreasing share of soybean production. Specifically, the yield per unit area of soybean in China is substantially lower than that of other major crops, such as wheat, rice, and maize (Liu et al., 2021). In addition, as China shifts from domestic cultivation of soybeans to importation, a considerable amount of arable land is being repurposed for the cultivation of other, more productive crops (Cui and Shoemaker, 2018).

These points have been comprehensively addressed and supplemented with supporting literature in the revised manuscript:

"Given the rapid growth of demand and the shortages of domestic supply due to low yield and low self-sufficiency, mapping soybean planting areas across China is crucial for sustainable soybean production and management (Cui and Shoemaker, 2018; Liu et al., 2021)."

Reference:

Cui, K. and Shoemaker, S. P.: A look at food security in China, npj Sci. Food, 2, 4, https://doi.org/10.1038/s41538-018-0012-x, 2018.

Liu, Z., Ying, H., Chen, M., Bai, J., Xue, Y., Yin, Y., Batchelor, W. D., Yang, Y., Bai, Z., Du, M., Guo, Y., Zhang, Q., Cui, Z., Zhang, F., and Dou, Z.: Optimization of China's maize and soy production can ensure feed sufficiency at lower nitrogen and carbon footprints, Nat. Food, 2, 426–433, https://doi.org/10.1038/s43016-021-00300-1, 2021.

Comment 2: L46: Please add references to previous studies.

Yes, we have followed you to add references here.

"Soybean planting area in some regions of China was mapped in previous studies (You et al., 2021; Huang et al., 2022; Chen et al., 2023), but long-term soybean maps over all major producing areas in China have not been available."

Comment 3: L59-62: I would suggest revising the statements as "the previous studies made laudable efforts to craft a comprehensive national maize-soybean map for China in 2019 by combining field data and regression estimators (Li et al., 2023). Nonetheless, these studies were confined to specific regions or a single year, despite prior attempts to accurately map soybean cultivation areas."

Thank you very much for your instructive comments. Your suggestion has indeed made the statement clearer and more logically coherent. We have revised the sentence as you suggested.

Comment 4: L64-70: to me, this is not "generally" way of categorizing remote sensing classification methods. Supervised and unsupervised are the widely accepted categories. I would suggest authors revise the paragraph, link the specific classification method mentioned in L71-78 to each category, and discuss the pros and cons.

Yes, we have reorganized the previous researches and divided the commonly used remote sensing-based crop classification methods into four categories. In addition to the supervised and unsupervised classification in machine learning that you mentioned, considering that threshold segmentation based on prior knowledge and new composite index methods based on feature bands are two other methods of crop mapping, we have summarized the methods into four types. Method 5 in the original text has been incorporated into supervised classification. Additionally, we revised the corresponding section, as well as discussing the advantages and disadvantages of each method: "Mapping crops by remote sensing can be categorized into four methods : 1) supervision classification based on a large number of field samples or high quality training labels (Song et al., 2017; You et al., 2021; Shangguan et al., 2022; Li et al., 2023); 2) developing some composite indexes based on the feature bands and determining the binary classification using appropriate threshold value (Huang et al., 2022; Chen et al., 2023; Zhou et al., 2023); 3) threshold segmentation based on prior knowledge such as phenology or spectra (Zhong et al., 2016); 4) combining unsupervised classification with post-classification (Wang et al., 2019; You et al., 2023). Supervision classification methods relied on ground samples heavily, while the 2$^{nd}$ and 3$^{rd}$ methods are both based on reliable and accurate thresholds. However, mapping soybean by these methods was mainly applied in small areas, very few covering over a larger region. Because of sufficient field samples, supervision classification can achieve maps with a higher accuracy, which is relatively mature method used widely. However, collecting sufficient field samples is extremely time, money, and labor costly, and unsuitable for long-term years and over larger areas (Luo et al., 2022). Furthermore, the threshold-based methods (the 2$^{nd}$ and 3$^{rd}$) have been applied into large areas, however, determining the thresholds will inevitably bring significant uncertainty, especially for the areas with high heterogeneity in climate, environment, and planting patterns. Thus, reproducibility of these methods is low, further hindering their application across diverse geographic areas. As for mapping soybean, it is still a big challenge due to their similar growth characteristics with many other summer crops (Wang et al., 2020; Di Tommaso et al., 2021). The thresholds that work well in some areas did not perform well in other areas (Graesser and Ramankutty, 2017; Guo et al., 2018). These limitations restrict accurate soybean maps available, especially over large regions in China."

Reference:

Chen, H., Li, H., Liu, Z., Zhang, C., Zhang, S., and Atkinson, P. M.: A novel Greenness and Water Content Composite Index (GWCCI) for soybean mapping from single remotely sensed multispectral images, Remote Sens. Environ., 295, 113679, https://doi.org/10.1016/j.rse.2023.113679, 2023.

Di Tommaso, S., Wang, S., and Lobell, D. B.: Combining GEDI and Sentinel-2 for wall-to-wall mapping of tall and short crops, Environ. Res. Lett., 16, 125002, https://doi.org/10.1088/1748-9326/ac358c, 2021.

Graesser, J. and Ramankutty, N.: Detection of cropland field parcels from Landsat imagery, Remote Sens. Environ., 201, 165–180, https://doi.org/10.1016/j.rse.2017.08.027, 2017.

Guo, W., Ren, J., Liu, X., Chen, Z., Wu, S., and Pan, H.: Winter wheat mapping with globally optimized threshold under total

quantity constraint of statistical data, Journal of Remote Sensing, 22, 1023–1041, https://doi.org/10.11834/jrs.20187468, 2018.

Huang, Y., Qiu, B., Chen, C., Zhu, X., Wu, W., Jiang, F., Lin, D., and Peng, Y.: Automated soybean mapping based on canopy water content and chlorophyll content using Sentinel-2 images, Int. J. Appl. Earth Obs., 109, 102801, https://doi.org/10.1016/j.jag.2022.102801, 2022.

Li, H., Song, X.-P., Hansen, M. C., Becker-Reshef, I., Adusei, B., Pickering, J., Wang, L., Wang, L., Lin, Z., Zalles, V., Potapov, P., Stehman, S. V., and Justice, C.: Development of a 10-m resolution maize and soybean map over China: Matching satellite-based crop classification with sample-based area estimation, Remote Sens. Environ., 294, 113623, https://doi.org/10.1016/j.rse.2023.113623, 2023.

Luo, Y., Zhang, Z., Zhang, L., Han, J., Cao, J., and Zhang, J.: Developing High-Resolution Crop Maps for Major Crops in the European Union Based on Transductive Transfer Learning and Limited Ground Data, Remote Sens., 14, 1809, https://doi.org/10.3390/rs14081809, 2022.

Shangguan, Y., Li, X., Lin, Y., Deng, J., and Yu, L.: Mapping spatial-temporal nationwide soybean planting area in Argentina using Google Earth Engine, Int. J. Remote Sens., 43, 1724–1748, https://doi.org/10.1080/01431161.2022.2049913, 2022.

Song, X.-P., Potapov, P. V., Krylov, A., King, L., Di Bella, C. M., Hudson, A., Khan, A., Adusei, B., Stehman, S. V., and Hansen, M. C.: National-scale soybean mapping and area estimation in the United States using medium resolution satellite imagery and field survey, Remote Sens. Environ., 190, 383–395, https://doi.org/10.1016/j.rse.2017.01.008, 2017.

Wang, S., Azzari, G., and Lobell, D. B.: Crop type mapping without field-level labels: Random forest transfer and unsupervised clustering techniques, Remote Sens. Environ., 222, 303–317, https://doi.org/10.1016/j.rse.2018.12.026, 2019.

Wang, S., Di Tommaso, S., Deines, J. M., and Lobell, D. B.: Mapping twenty years of corn and soybean across the US Midwest using the Landsat archive, Sci. Data, 7, 307, https://doi.org/10.1038/s41597-020-00646-4, 2020.

You, N., Dong, J., Huang, J., Du, G., Zhang, G., He, Y., Yang, T., Di, Y., and Xiao, X.: The 10-m crop type maps in Northeast China during 2017–2019, Sci. Data, 8, 41, https://doi.org/10.1038/s41597-021-00827-9, 2021.

You, N., Dong, J., Li, J., Huang, J., and Jin, Z.: Rapid early-season maize mapping without crop labels, Remote Sens. Environ., 290, 113496, https://doi.org/10.1016/j.rse.2023.113496, 2023.

Zhong, L., Hu, L., Yu, L., Gong, P., and Biging, G. S.: Automated mapping of soybean and corn using phenology, ISPRS J. Photogramm. Remote Sens., 119, 151–164, https://doi.org/10.1016/j.isprsjprs.2016.05.014, 2016.

Zhou, W., Wei, H., Chen, Y., Zhang, X., Hu, J., Cai, Z., Yang, J., Hu, Q., Xiong, H., Yin, G., and Xu, B.: Monitoring intra-annual and interannual variability in spatial distribution of plastic-mulched citrus in cloudy and rainy areas using multisource remote sensing data, European Journal of Agronomy, 151, 126981, https://doi.org/10.1016/j.eja.2023.126981, 2023.

Comment 5: L121: Please justify the impact of using TOA reflectance, rather than surface reflectance, on classification results.

During using Sentinel-2 imagery in our study, we encountered difficult with the L2A product on the GEE platform in terms of temporal coverage in China. Taking the Northeast as an example, the L2A data was only available after December 2018, whereas the L1C product offered complete coverage from 2017 onwards. Consequently, for crop mapping prior to 2019, L2A was not a viable option. To be consistency, we opted for the L1C product for mapping soybean.

Furthermore, to ensure the reliability of L1C product for classification, we analyzed spectral and vegetation indices time series from field samples in Daqing, Heilongjiang Province, for both L1C and L2A products in 2019 (Figures 1-2). The difference between two spectral profiles is minimal. More importantly, the L1C-based spectral and vegetation indices also demonstrated effective separability between soybeans and other crops. Thus, to preserve the temporal integrity without compromising classification accuracy, we chose Sentinel's L1C (TOA), rather than L2A (SR).

In section 2.2.1 of the revised manuscript, we have added an explanation for our choose for L1C

instead of L2A.

"... last access: September 2023). Because of the longer temporal coverage of Sentinel-2 Level-1C TOA reflectance data, and the nearly identical spectral profile time series extracted from both products demonstrating that TOA images can equally full fill crop classification requirements, we opt for using L1C products instead of L2A (You and Dong, 2020; Han et al., 2021; Luo et al., 2022)."

Reference:

Han, J., Zhang, Z., Luo, Y., Cao, J., Zhang, L., Zhang, J., and Li, Z.: The RapeseedMap10 database: annual maps of rapeseed at a spatial resolution of 10 m based on multi-source data, Earth Syst. Sci. Data, 13, 2857–2874, https://doi.org/10.5194/essd-13-2857-2021, 2021.

Luo, Y., Zhang, Z., Zhang, L., Han, J., Cao, J., and Zhang, J.: Developing High-Resolution Crop Maps for Major Crops in the European Union Based on Transductive Transfer Learning and Limited Ground Data, Remote Sens., 14, 1809, https://doi.org/10.3390/rs14081809, 2022.

You, N. and Dong, J.: Examining earliest identifiable timing of crops using all available Sentinel 1/2 imagery and Google Earth Engine, ISPRS J. Photogramm. Remote Sens., 161, 109–123, https://doi.org/10.1016/j.isprsjprs.2020.01.001, 2020.

[Figure]

**Figure 1. Temporal profiles of L2A products for major crops in Daqing, Heilongjiang based on ground samples.**

[Figure]

**Figure 2. Temporal profiles of L1C products for major crops in Daqing, Heilongjiang based on ground samples.**

Comment 6: L123: Depending on the platform/sensor used, red edge bands are also typical "traditional bands" in vegetation-related studies.

Thank you for pointing out the issue. Indeed, the red-edge bands have been deployed on various sensors and have become primary application bands. We have removed the expressions that could cause ambiguity in the revised manuscript:

"The red-edge bands and shortwave infrared bands equipped with sentinel-2 play a great role in enhancing the accuracy of crop classification."

Comment 7: L135: Please specify what is the "gaps"? If it is related to crop growth, how the "average" procedure was conducted?

Specifically, 'gaps' means the missing phenological observations in a certain year at some agricultural meteorological stations we collected. For the missing values, we inserted averages of the observations from the nearest years before and after the missing year. For example, if the flowering date in 2017 was missing, we inserted the average of flowering dates in 2016 and 2018 at that station as a substitute. We have rewritten and clarified this issue in section 2.2.2 of the revised manuscript:

"In cases of missing observation for a specific year, we inserted the average of two closest observations before and after the year. For instance, if there was missing data of flowering date in 2017, we filled it with the average of flowering records in 2016 and 2018 at the same station. "

Comment 8: L189: it seems the purpose of this paragraph is to provide an overview of the method. The details regarding the "soybean mapping" can be merged with the sections below.

Thank you for the constructive suggestion. We have streamlined the description of soybean mapping methodology in this paragraph, and merged the details with the following sections as you suggested. Such revision really enhances the clarity and conciseness of the methodology section.

Comment 9: L198-L200: I recon this is also the step that deals with the data gaps due to cloud? Please add more details regarding the method incorporated (e.g. moving window size etc?) if possible.

Yes, the time series reconstruction is carried out to simultaneously fill data gaps caused by cloud removal and smooth some anomalies. In order to obtain 10-day composite time series, as well as considering the revisit cycle of Sentinel-2 and computational efficiency, we set the half-window size to 10 days. We have added the details in the revised manuscript:

"To fill the data gaps caused by cloud removal and smooth anomalies, Sentinel-2 time series was reconstructed by moving median composite method, resulting in a 10-day interval composite time series. We set the half-window size for the moving median methods to 10 days considering the 5-day revisit cycle of Sentinel-2 and computational efficiency. "

Comment 10: L203: no-cropland --> non-cropland

Thank you for pointing out this mistake, we have corrected it throughout the manuscript.

Comment 11: L204: you might need to define the "starting and ending dates of the growing season" first.

Following your suggestions, we have defined the sowing dates recorded at the nearest AMS as the starting dates of growing season, and the harvesting dates as ending dates. This has been clarified in section 2.3.2 "(1) Potential area identification" in our revised manuscript:

"To minimize the impact from non-croplands, we firstly determine the potential cropping areas by masking GLAD cropland layer over study area. Sentinel-2 images within growing season were extracted by taking the sowing date and harvesting date recorded at the nearest agricultural meteorological station (AMS) as the starting and ending dates of the growing season, respectively."

Comment 12: L206: Please provide the full name for EVI first. And, revise the sentence slightly, "… we masked out the pixels with maximum EVI less than 0.4 during the growing seasons". Please also justify how the threshold (0.4) for fallow land was determined.

Thank you for your suggestion. We have followed you to provide the full name of EVI in the revised manuscript. We identified the pixels with maximum EVI values < 0.4 as fallow land because the maximum EVI values for crops are all > 0.4 (except for a few outliers) (Figure 3) based on all ground samples in 2019 (Figure 3). Thus, using 0.4 as a threshold allows us to strictly remove fallow land (Li et al., 2014). We have provided additional explanations for the threshold choice in the revised manuscript:

"Based on the cropland extracted, we filtered the pixels exhibiting an Enhanced Vegetation Index (EVI) maximum value during the growing season greater than 0.4 to remove fallow land, because ground samples and previous studies showed that nearly all crops had maximum EVI values above 0.4. (Li et al., 2014)."

Reference:

Li, L., Friedl, M. A., Xin, Q., Gray, J., Pan, Y., and Frolking, S.: Mapping Crop Cycles in China Using MODIS-EVI Time Series, Remote Sens., 6, 2473–2493, https://doi.org/10.3390/rs6032473, 2014.

[Figure]

**Figure 3. Box plot of the EVI maximum in 2019 based on ground samples.**

Comment 13: L296- : it is good that the authors noticed the large estimation uncertainties in small-planting regions (figure 4, and figures 5). It would help to justify why this happened by looking into several regions and checking the reasons.

Also, given the great similarities of maize and soybean index profiles (Figure 3), it is important to check whether the overestimated regions belong to maize crops? Since the classifiers are trained for individual regions, the authors might consider increasing the number of clusters for sparsely planting regions if maize is mixing with soybean due to their similarities? One potential way to check is to compare the ChinaSoyArea10m with the GLAD layer, especially the overestimating regions?

Yes, we compared ChinaSoyArea10m with the GLAD layer in the Shandong region, as the consistency between GLAD and statistics is higher there. Apart from the pixels consistently recognized as soybeans by both layers, some cornfields identified by the GLAD layer are classified into soybeans by ChinaSoyArea10m.

[Figure]

**Figure 4. Visual comparison of GLAD (a1-a2) and ChinaSoyArea10m (b1-b2) in China (a1-b1) and a typical area in Shandong province (a2-b2).**

In some provinces where we might overestimate soybean areas (e.g. Sichuan, Shaanxi, and Shanxi), GLAD significantly underestimated the soybean areas comparing with statistics (Figure 5). Therefore, it is very hard for us to determine which product is more accurate and reliable in such areas sparsely planted. We have discussed the uncertainties in details in the first paragraph of section 3.1.

"This uncertainty, particularly overestimation, could be caused by the low proportion of soybean cultivation. In areas where maize or other same-season crops are planted in a much larger proportion than soybeans, soybeans, as a less prevalent crop, pose a challenge for classifiers to distinctly recognize them as a separate category, resulting in clusters being identified as soybeans containing maize or other crops."

[Figure]

**Figure 5. Box plot of soybean areas of statistics and GLAD map in Sichuan, Shaanxi, and Shanxi.**

Comment 14: L315: to me, "became higher and higher" is not a scientific way to describe the trend here. Please consider "increased" or similar terms for the statement if it is a critical finding.
Thanks! We have used 'increased' in the manuscript.

Comment 15: L353: It is good to see the authors outline the limitations of the proposed method in regard to its sensitivity to data availability and applicability in sparsely planted regions. It would be good to have some insights into the advantages of the method compared to the mentioned GLAD and CDL products and promote its applications in some suggested circumstances.
Thank you for your suggestion. We have added a section on our advantages and potential applicability. We highlighted the strengths of our method: its independence from extensive requirements for samples, and its capability for rapid mapping in other regions along with excellent spatial scalability. Unlike the previous products relied on extensive sample points for supervised classification, our approach could be applied into other major soybean-producing areas with simple inputs.

**4.1 Our advantages and potential applicability**

The methodology developed for identifying soybean planting areas indicate several notable strengths that make it an attractive option for wide application. Firstly, it operates independently, without extensive ground samples required. The conventional approaches depend on quantities of observational data, with much money, time, and labor consumed. In contrast, our strategy leverages a specific, pre-existing set of sample points to discern soybean characteristics. This approach can accurately map annual dynamics of soybean planting areas without updated requirement in annual samples. Consequently, this method significantly weakens limitations in crop classification during years without specific samples, enabling crop mapping consistently and continually.
Another key advantage of our spectra-phenology integration approach is its rapid applicability over wide areas, coupled with excellent spatial scalability. The only inputs required for our mapping technique are the phenological information of soybeans and other primary crops during the same growing season in the target area. This allows to classify crop swiftly and efficiently without additional needs for background knowledge or setting complex thresholds. The input of phenological information in each prefecture ensured that the zonal adaptive soybean growth status across various regions could be taken into account in classification. Given that soybeans exhibit similar spectral characteristics during identical phenological stages from the same sub-zone in the study area, our method utilizes standard soybean sample curves from various regions to identify clusters most likely representing soybeans. This innovative approach ensures our methodology's applicability across major soybean-producing regions, showcasing its potential for broader implementation.

---

## Author Comment (AC2)

**Response to reviewer #2:**

Many thanks for your thoughtful and valuable comments and suggestions, which are very helpful in improving our manuscript. We have conducted substantial new experiments and analyses to ensure that the study is more comprehensive and rigorous, and our maps are more reliable. Our responses to the comments point-by-point are included below in blue. The corresponding changes in the revised manuscript are shown in purple.

**General comment:** The ms employed two steps method to map soybean at large scale in China for 2017-2021. While the topic and the generated dataset have great potential to benefit the agriculture community in both research and operational monitoring aspects, there are some major flaws that need to be addressed to enhance the scientific soundness of the paper and the reliability of the data. The authors listed three objectives. The new data product of soybean maps was generated and openly shared to address the third objective. However, the first two objectives have not been thoroughly investigated. Further examination is required to test the method's robustness in extracting soybean fields across different regions. Although the nationwide validation using ground samples shows generally acceptable accuracy, the variations in accuracy among regions need to be illustrated. This can be easily done as the classification was applied at the prefecture level. Additionally, the accuracy in low soybean growing regions should be specified. The proposed method appears to be ineffective in accurately extracting soybean fields and lacks effectiveness in non-soybean producing provinces. In this case, it may not be meaningful to generate soybean map at a national scale while most non-producing provinces presents unreliable results. Additionally, the validation process is questionable since the data used to determine soybean clusters was also used in the validation.

**Reply:** Thank you for such expert questions, which do help us to deepen our study. To respond fully the above problems, we have correspondingly revised the manuscript.

(1) We illustrated the accuracy assessment results based on statistics and samples in each sub-zone, and explained the differences among regions.

➤ **The variations in accuracy among sub-zones based on statistics validation:**

"The mapping accuracy in Zone I closely matched county-level statistics, showing high consistency ($R^2$=0.86). Zones II-IV also demonstrated reasonable agreement ($R^2$=0.50~0.69), despite relatively lower accuracy due to the scarcer planted areas (Fig. S5). No significant trend deviation from statistics was indicated for the mapping area in Zone I, with slight overestimates for Zone II and III, and underestimates for Zone IV (Fig. S5). These accuracy variations are acceptable, given the challenges in accurately identifying soybeans in regions in less prevalent regions. Specifically, maize is more dominant than soybeans in Zone II, while Zone III is characterized by diverse crops and complex planting patterns. Underestimation in Zone IV is possibly due to fewer clear observations in the southwest. Nevertheless, the overall accuracy across the zones is acceptable."

[Figure]

**Figure S5. Comparison of soybean areas with county-level statistics in (a) Zone I, (b) Zone II, (c) Zone III, and (d) Zone IV in 2017 and 2018.**

➢ **The variations in accuracy among sub-zones based on samples validation:**

"The overall accuracy for each sub-zone in 2019 varied from 83.58% to 90.67% (Table S1). Specifically, Zone I demonstrated the highest producer's accuracy for soybean at 88.31%, aligning with its high consistency with statistics. Zone III achieved the highest overall accuracy at 90.67%, attributed to its superior user's accuracy for soybean, indicating fewer misclassifications, and effective differentiation from non-soybean crops (Table S1). The producer's accuracy in Zone IV was relatively lower at 63.89%, possibly due to the limited samples, high heterogeneity, and fewer clear observations (Table S1)."

**Table S1. Confusion matrix of the soybean maps in each sub-zone in 2019.**

|   | Reference | Map Soybean | Non-Soybean | Producer's Accuracy | User's Accuracy | F1 Score | Overall Accuracy |
|---|---|---|---|---|---|---|---|
| I | Soybean | 922 | 122 | 88.31% | 81.09% | 0.85 | 87.12% |
|   | Non-Soybean | 215 | 1358 | 86.33% | 91.76% | 0.89 | |
| II | Soybean | 233 | 74 | 75.90% | 86.30% | 0.81 | 83.58% |
|   | Non-Soybean | 37 | 332 | 89.97% | 81.77% | 0.86 | |
| III | Soybean | 101 | 26 | 79.53% | 98.06% | 0.88 | 90.67% |
|   | Non-Soybean | 2 | 171 | 98.84% | 86.80% | 0.92 | |
| IV | Soybean | 23 | 13 | 63.89% | 92.00% | 0.75 | 87.18% |
|   | Non-Soybean | 2 | 79 | 97.53% | 85.87% | 0.91 | |

(2) We further discussed the mapping accuracy in the areas planted sparely. Although the verification accuracy there are not as good as those in main producing areas, its accuracy is still acceptable. The effectiveness in key regions indicates the potential application of our method, and the mapping nationwide provides insights for differentiated policy formulation across regions. Therefore, it is meaningful to map soybean area on a national scale (see Reply for Comment 8 for more details). We have explained the accuracy differences in lower soybean producing areas (see (1) above), and added the reasons into the discussion section for the higher uncertainty there and possible solutions in the future (see Reply for Comment 9).

(3) We updated the validation results in the manuscript (see Reply for Comment 7). All points are divided according to the ratio of 3:7, which are used to determine the standard curves of each sub-zone and verify the mapping accuracy to ensure scientific and independent verification.

**Specific comments:**

**Comment 1:** Line 29, Cropland Data Layer or Crop Data Layer? The existing maps are described as crop type maps not cropland maps.

**Reply:** Thank you pointing out the issue. The full name of CDL was not specified in the publication, and we have changed its full name to Crop Data Layer since this dataset represents crop types.

**Comment 2:** In second paragraph of introduction section, it is recommended to specify the research study areas for each citation when highlighting their work. For example, line 52 to 55. I thought the research generated 20-years maize-soybean maps for whole China but it is not.

**Reply:** Thank you for your valuable comments. We have checked all the citations in the second paragraph and have specified the studied areas that were not clearly stated in the revised manuscript: "More recently, 20-year soybean-corn maps with 30 m resolution across the US Midwest have been generated by collecting a large number of samples and using green chlorophyll vegetation index (GCVI) time series features, which is a large-scale, high-precision soybean mapping attempt (Wang et al., 2020)."

**Comment 3:** Line 145-147, does National Bureau of Statistics of China provide county and prefecture level data? How to you use national and provincial data to validate at county and prefecture level information?

**Reply:** We accessed to statistical yearbooks through the National Bureau of Statistics and obtained yearly county- and prefecture-level soybean planting area statistics from the yearbook of each county or province. We have declared this in the revised manuscript:
"...we utilized agricultural census data obtained from the statistical yearbook of each county or province by accessing National Bureau of Statistics of China (http://www.stats.gov.cn/, last accessed: June 2023)."
Accordingly, we used finer statistics at county- and prefecture- level for validation, rather than national and provincial data.

**Comment 4:** Line 215-217, how high the uncertainty resulting from the cloud cover or miss values during the proposed period?

**Reply:** Yes, it is really interesting to quantify how much of the uncretainty from the cloud cover or miss values during the proposed period (from 15 days before the podding date to 15 days after the full-seed date). We summarized the clear observations during the proposed period in each year (Figure 1) and found that few areas in Zones I-III and less than 10% of the areas in Zone IV had 0

or 1 clear observation in each year (except for 2017). Majority of areas have more than 1 clear observation during the proposed period, so the maximum EVI can be detected. Moreover, we tried to minimize the uncertainty by reconstructing the time series, and we took an example for details:

We selected the area with cloud cover during the proposed period in 2018 (Figure 2a). The Sentinel-2 median composite image showed that some parts of the land was bare soil, with the corresponding ground sample points marked as wheat (different from the growing period of soybean). We extracted the non-fallow areas (Figure 2b), seasonal crop areas (Figure 2c), and the difference of the two layers representing non-seasonal crop areas (Figure 2d). The removed plots correspond precisely to the wheat samples and the bare soil areas in Figure 2a. The extraction results show that even with substantial cloud cover during the proposed period, areas covered by clouds are not removed as non-seasonal crops because time series reconstruction minimize the impacts of cloud cover as much as possible. We have clarified this in section 2.3.2 "(1) Potential area identification" of the revised manuscript and discussed the uncertainty in section 4.

"The impact of cloud-covered pixels appearing in the proposed period is minimized since we have reconstructed the original EVI time series."

[Figure]

**Figure 1. The times of clear observations in proposed period by sub-zone in (a) 2017, (b) 2018, (c) 2019, (d) 2020, and (e) 2021.**

[Figure]

**Figure 2. Case of seasonal crop identification. (a) RGB composite image comprise red (Band 4), green (Band 3), and blue (Band 2) bands from Sentinel-2 median composite images during the proposed period; (b) areas of fallow land removed on the cropland layer; (c) areas of non-seasonal crop land removed on the areas corresponding to (b); (d) non-seasonal crop mask.**

**Comment 5:** Line 255-257 not clearly stated. Any quantitative information to determine whether crops are major ones or minor ones? It is problematic when statistical area of some crops in double cropping pattern, for example double rice.

**Reply:** Sorry for the ambiguity. We collected the statistical area for seasonal crops (including rice, maize, soybean, cotton, peanuts, sesame, sweet potato, and sorghum) in each prefecture in 2018. We defined "major crops" as those species cumulatively representing 95% of the whole seasonal cropping area, with an additional category for "other crops" to determine the number of clusters $k$. We have added the definition in section 2.3.2 "(3) Unsupervised learning":

"The classifier was trained individually in each prefecture based on the number of clusters $k$ input. The cluster number $k$ is defined as the number of "major crops" that constituting 95% of the total area of seasonal crops (including rice, maize, soybean, cotton, peanuts, sesame, sweet potato, and sorghum) according to prefecture-level statistics, plus one for "other crops"."

Double cropping pattern was mainly distributed in zone II and III, where soybean and other seasonal crops are planted in turn with winter crops (Yan et al., 2019). In particular, double-cropping rice is mainly planted in region III (Pan et al., 2021a). We collected statistical planting area of single-cropping rice and double-cropping late rice, treating them as two categories because of the different growing seasons.


**Reply:** The analysis showed the standard curves of soybean is very similar in a certain area during our studied years (Figure 3). Therefore, we determined the standard curves in each sub-zone, which can be applied into other years or the similar cropping systems. In provinces without ground samples, we manually select the reference points based on GLAD soybean-maize map (Figure S1). The criterions selected are: (1) located in large plots; (2) false color composite image (R: NIR, G: SWIR2, B: SWIR1) at the peak of growing season (Song et al., 2017; You and Dong, 2020); (3) phenological characteristics similar to local observations. The seasonal change of soybean for each zone does not vary from year to year based on our analysis (Figure 3), thus the characteristic curves in 2019 were taken as uniform standard.

[Figure]

**Figure S1. Spatial distribution of ground samples and reference points.**

[Figure]

**Figure 3. EVI of soybean ground samples in a same area from 2017 to 2019.**

We updated the description of data sources in section 2.2.4 "Census data and ground samples" of the manuscript:

"We used both ground samples and reference points based on available datasets to determine soybean standard curves and assess the reliability of the soybean maps (Fig. S1). All points were randomly divided in a 3:7 ratio for standard curve calculation and accuracy validation, respectively. We collected ground samples from field surveys from 2017 to 2019 in Heilongjiang (HLJ), Inner Mongolia (NMG), Anhui (AH), Henan (HN), and Jilin (JL), which account for more than 70% of the country's total soybean planting area (Table 1). Crop types (soybean, maize, rice, wheat, others) and other land cover types were recorded. To ensure the impartiality of verification results, we only selected crop samples for validation. In provinces without ground samples, we manually selected reference points on large soybean plots based on GLAD (https://glad.earthengine.app/view/china-crop-map, last access: March 2024) soybean layer.

The criterions selected are: (1) located in large plots; (2) false color composite image (R: NIR, G: SWIR2, B: SWIR1) at the peak of growing season (Song et al., 2017; You and Dong, 2020); (3) phenological characteristics similar to local observations. Additionally, the reference points of maize, single-cropping rice and double-cropping rice in 2019 were selected based on GLAD maize layer, high resolution single-season rice map (https://doi.org/10.57760/sciencedb.06963, last access: March 2024), and double-season rice map (https://doi.org/10.12199/nesdc.ecodb.rs.2022.012, last access: March 2024) with the same principle to explore the spectral characteristics of crops in each sub-zone of the studied areas. The overall accuracy of all available maps in 2019 is above 85% (Pan et al., 2021; Li et al., 2023; Shen et al., 2023)."


**Comment 7:** Those ground samples were used in both cluster assignments and validation. Scientifically, independent validation shall be applied.

**Reply:** Thank you very much for your instructive comments. Yes, the ground samples used to cluster assignment and validation should be separated. The samples were randomly divided according to the ratio of 3:7 for standard curves calculation and accuracy validation respectively. The average characteristic curves of the points drawn according to the 30% ratio are almost indistinguishable from the average curves of all points. The accuracy verification results of 70% sample points are updated as follows:

Table 2. Confusion matrix of the soybean maps during 2017-2019.

|  | Reference | Map Soybean | Non-Soybean | Producer's Accuracy | User's Accuracy | F1 Score | Overall Accuracy |
|---|---|---|---|---|---|---|---|
| 2017 | Soybean | 679 | 352 | 65.86% | 72.47% | 0.69 | 77.08% |
|  | Non-Soybean | 258 | 1372 | 84.17% | 79.58% | 0.82 |  |
| 2018 | Soybean | 799 | 246 | 76.46% | 74.19% | 0.75 | 85.16% |
|  | Non-Soybean | 278 | 2208 | 88.82% | 89.98% | 0.89 |  |
| 2019* | Soybean | 1279 | 235 | 84.48% | 83.32% | 0.84 | 86.77% |
|  | Non-Soybean | 256 | 1940 | 88.34% | 89.20% | 0.89 |  |

* Including ground samples and nationwide reference points based on existing datasets.

**Comment 8:** According to the validation in 3.1, it seems that the mapping accuracy is much lower in counties with less soybean area at both county and prefecture level. This does not surprise me due to the combined resolution bias and the algorithm uncertainties. This raise up another question, is it meaningful to generate soybean maps at almost whole national scale?

**Reply:** We recognize the limitations in non-main producing areas, yet it still makes sense to conduct a nationwide soybean areas extraction based on the following reasons.

(1) The validation accuracy of each sub-zone is acceptable. By integrating ground samples with reference points from existing datasets, we have supplemented the validation results for each sub-zone in 2019 (Table S1). The OA values reached 83.58% ~ 90.67%, with only producer's accuracy of Zone IV was relatively lower at 63.89%, which may be due to the limited samples, high heterogeneity, and fewer clear observations.

(2) Errors can be caused by multiple sources. In scarcer planted areas, the misclassification or omissions show a greater impact on the results. In addition to classifier constraints, the mapping area aggregate based on pixel counts may also lead to errors due to mixed pixels and resolution limitations in scattered planting areas.

(3) The method still has great application potential in main soybean-producing areas. Our mapping results showed robust interannual high $R^2$ and OA, demonstrating that mapping is valid and reliable in key areas. Therefore, the method can be used in other major soybean production areas in the world and has great practical application potential.

(4) Conducting nationwide soybean mapping is beneficial, providing insights for practical use and method improvement in future. Even with uncertainties in non-main producing areas, identifying possible soybean distributions can provide references for local agricultural monitoring and policy making. In addition, understanding soybean planting patterns in different regions and further exploring the spatial differences in production are very beneficial to soybean production adjustment at the national level. The method can be further improved by collecting more detailed crop pattern information, classifying on finer scale, improving classification algorithms, and integrating more various data sources to identify soybean plots in these regions (see Reply for Comment 9 for more details).

**Comment 9:** The discussion needs significant improvements. The author discussed the limitations of the research while ignoring the strong points of the research. Also, the uncertainty of the classification at small-scale soybean cultivation areas shall be addressed from a more theoretical way.

**Reply:** Thank you for your instructive suggestion. We added "4.1 Our advantages and potential applicability" to the discussion. We highlighted the strengths of our method: its independence from extensive requirements for samples, and its capability for rapid mapping in other regions along with excellent spatial scalability. Unlike the previous products relied on extensive samples for supervised classification, our approach could be applied into other major soybean-producing areas with simple inputs.

**4.1 Our advantages and potential applicability**

[revised manuscript text omitted]

**Comment 10:** The ms does not consider the soybean-maize intercropping systems in part of China.
**Reply:** We have considered all cropping patterns related with soybean as possible. For example, we have identified the banded areas planted by soybean as shown in Figure 4. Soybean and other crops are interplanted with the narrowest strip width detectable at about 2-3 pixels, equivalent to 20-30

meters. However, if the strip widths in soybean-maize intercropping system are less than a meter (Yang et al., 2014; Du et al., 2018), the 10-meter resolution provided by Sentinel-2 imagery will fail in capturing these planting stripes due to mixed pixels problems. With the help of higher resolution remote sensing data, such as sub-meter level satellite images or local unmanned aerial vehicle (UAV) images, such dense intercropping systems will be identified more accurately. We added discussion to state the uncertainty caused by this agricultural pattern:

"The growth periods of soybean, peanut, potato, and maize are similar, dominantly indicated by a mixed planting pattern, which has contributed to the low accuracy of non-main soybean producing areas in southern China (Liu et al., 2020). Additionally, soybeans are intercropped with maize or other crops in some areas, where the strip width is less one meter (Yang et al., 2014; Du et al., 2018). This planting pattern will introduce the mixed pixels problem as well under the background of 10 m resolution crop mapping."

[Figure]

**Figure 4. The case of soybean banded planting pattern in northeast China.**

---

## Author Comment (AC3)

**Response to reviewer #3:**

Many thanks for your thoughtful and valuable comments and suggestions, which are very helpful in improving our manuscript. We have conducted substantial new experiments and analyses to ensure that the study is more comprehensive and rigorous, and our maps are more reliable. Our responses to the comments point-by-point are included below in blue. The corresponding changes in the revised manuscript are shown in purple.

**General comment:** This manuscript developed a phenological- and pixel-based soybean area mapping (PPS) method to identify soybean on a large scale and generated a dataset of soybean planting areas across China. The topic is significant for sustainable soybean production and management. However, the proposed methodology lacks notable innovation when compared to prior studies. Given the intricate spectral variations within soybeans and the fragmented nature of agricultural landscapes across China, the presented method fails to demonstrate its robustness across diverse regions and time periods, therefore raising concerns about the reliability of the resulting soybean map. Furthermore, certain descriptions of the proposed method lack essential details and specific contents are not easy to follow. Below, I have provided several detailed comments:

**Reply:** Thank you for the expert questions, which do encourage us to comb our method and add more detailed experiments to demonstrate the robustness of our approach and the reliability of the resulting soybean maps. We explained all these details by three sections (innovation, accuracy, and methods' details).

(1) Innovation: We firstly obtained the characteristic spectra and growth curves of soybean in different areas during the key observed growth periods, and then trained local unsupervised classifiers to self-adapt to cross-regional growth variability, which have avoided huge requirement for extensive ground samples. The Regional Adaptation Spectra-Phenology Integration (RASP) framework proposed is novel and can be repeatable into other major areas planted by soybean with simple inputs, providing a solution for mapping annual soybean dynamics with a higher resolution (please see the details in the revised method section of the new edition from line 231-312 in page 11-15).

(2) Accuracy: In our results of accuracy assessment, we have used so many data available from different years to verify the reliability of our soybean maps by several methods, including visual comparison, comparing soybean areas retrieved with county- and prefecture- statistical books, and point verification with confusion matrix separately by sub-zone. Very few previous studies have assessed comprehensively maps' accuracy by the three methods. To demonstrate the robustness of mapping across diverse regions, we added the accuracy results of statistical data verification and point verification in each sub-zone.

➢ **The variations in accuracy among sub-zones based on statistics validation:**

"The mapping accuracy in Zone I closely matched county-level statistics, showing high consistency ($R^2$=0.86). Zones II-IV also demonstrated reasonable agreement ($R^2$=0.50~0.69), despite relatively lower accuracy due to the scarcer planted areas (Fig. S5). No significant trend deviation from statistics was indicated for the mapping area in Zone I, with slight overestimates for Zone II and III, and underestimates for Zone IV (Fig. S5). These accuracy variations are acceptable, given the challenges in accurately identifying soybeans in regions where they are planted less prevalently. Specifically, maize is more dominant than soybeans in Zone II, while Zone III is characterized by diverse crops and complex planting patterns. Underestimation in Zone IV is possibly due to fewer clear observations in the southwest.

Nevertheless, the overall accuracy across the zones is acceptable."

[Figure]

**Figure S5. Comparison of soybean areas with county-level statistics in (a) Zone I, (b) Zone II, (c) Zone III, and (d) Zone IV in 2017 and 2018.**

➤ **The variations in accuracy among sub-zones based on samples validation:**

"The overall accuracy for each sub-zone in 2019 varied from 83.58% to 90.67% (Table S1). Specifically, Zone I demonstrated the highest producer's accuracy for soybean at 88.31%, aligning with its high consistency with statistics. Zone III achieved the highest overall accuracy at 90.67%, attributed to its superior user's accuracy for soybean, indicating fewer misclassifications, and effective differentiation from non-soybean crops (Table S1). The producer's accuracy in Zone IV was relatively lower at 63.89%, possibly due to the limited samples, high heterogeneity, and fewer clear observations (Table S1)."

**Table S1. Confusion matrix of the soybean maps in each sub-zone in 2019.**

|  | Reference | Map | | Producer's | User's | F1 | Overall |
|---|---|---|---|---|---|---|---|
|  |  | Soybean | Non-Soybean | Accuracy | Accuracy | Score | Accuracy |
| I | Soybean | 922 | 122 | 88.31% | 81.09% | 0.85 | 87.12% |
|  | Non-Soybean | 215 | 1358 | 86.33% | 91.76% | 0.89 |  |
| II | Soybean | 233 | 74 | 75.90% | 86.30% | 0.81 | 83.58% |
|  | Non-Soybean | 37 | 332 | 89.97% | 81.77% | 0.86 |  |
| III | Soybean | 101 | 26 | 79.53% | 98.06% | 0.88 | 90.67% |
|  | Non-Soybean | 2 | 171 | 98.84% | 86.80% | 0.92 |  |
| IV | Soybean | 23 | 13 | 63.89% | 92.00% | 0.75 | 87.18% |
|  | Non-Soybean | 2 | 79 | 97.53% | 85.87% | 0.91 |  |

(3) Method descriptions in details: We have added the necessary details to the method we proposed in method section of our MS. To fully and positively respond all your valuable and suggestive comments, we also further listed them point by point in the follows.

**Specific comments:**

**Comment 1:** Line83-93: The expression and logic are not clear. I suggest that the authors reorganize "method (5)" to emphasize its key theory, advantages and disadvantages. Additionally, Line93-98 should be revised to describe the fundamental theory and performance of those method proposed by prior researchers. Furthermore, in Introduction section, the authors didn't introduce the fundamental concept behind the proposed method, nor highlighted the current issues faced by previous efforts in large-scale soybean mapping.

**Reply:** Thank you for your instructive suggestion. We have followed you to modify all these sentences thoroughly in the Introduction section:

(1) We have reorganized the previous researches and divided the remote sensing-based crop classification methods used widely into four categories. Method 5 in the original text has been incorporated into supervised classification. Additionally, we revised the corresponding section, as well as discussing the advantages and disadvantages of each method:

"Mapping crops by remote sensing can be categorized into four methods : 1) supervision classification based on a large number of field samples or high quality training labels (Song et al., 2017; You et al., 2021; Shangguan et al., 2022; Li et al., 2023); 2) developing some composite indexes based on the feature bands and determining the binary classification using appropriate thresholds (Huang et al., 2022; Chen et al., 2023; Zhou et al., 2023); 3) threshold segmentation based on prior knowledge such as phenology or spectra (Zhong et al., 2016); 4) combining unsupervised classification with cluster assignment (Wang et al., 2019; You et al., 2023). Supervision classification methods relied on ground samples heavily, while the 2$^{nd}$ and 3$^{rd}$ methods are both based on reliable and accurate thresholds. However, mapping soybean by these methods was mainly applied in small areas, very few covering over a larger region. Because of sufficient field samples, supervision classification can achieve maps with a higher accuracy, which is relatively mature method used widely. However, collecting sufficient field samples is extremely time, money, and labor consumed, and unsuitable for long-term years over larger areas (Luo et al., 2022). Furthermore, the threshold-based methods (the 2$^{nd}$ and 3$^{rd}$) have been applied into large areas, however, determining the thresholds will inevitably bring significant uncertainty, especially for the areas with high heterogeneity in climate, environment, and planting patterns. Thus, these methods show low reproducibility, further hindering their application across diverse geographic areas. As for mapping soybean, it is still a big challenge due to their similar growth characteristics with many other summer crops (Wang et al., 2020; Di Tommaso et al., 2021). The thresholds that work well in some areas did not perform well in other areas (Graesser and Ramankutty, 2017; Guo et al., 2018). These limitations restrict accurate soybean maps available, especially over large regions in China."

(2) We have described the fundamental theory and the performance of prior researches mentioned in the original line 93-98 as you suggested:

[revised manuscript text omitted]

**Comment 2:** Fig.1 shows that there are more soybean agrometeorological observation stations in Jiangxi Province than in Sichuan Province. So, why does the study area not include regions in South China, especially prefectures in the Jiangxi Province?

**Reply:** Yes, more soybean AMSs are located in Jiangxi Province, but we did not retrieve soybean areas there because of the quality limitations of Sentinel images available nowadays. Moreover, soybean planted in Southern China are generally scattered in fragmented and more complicated fields. It will be a very big challenge for smoothly selecting the specific features of a certain minor crop among many dominant crops. We excluded Southern China, including Jiangxi province, considering the above difficulties and their minor roles relative to the overall soybean production in China. All our responses to this comment are showed specifically as follows:

(1) According to the provincial statistics, the soybean planting area of the top 13 provinces accounts for over 90% of the whole national production, with only below 10% from other provinces. Therefore, despite of phenological observations available, we excluded the province from our analyses because of their minimal contribution.

(2) In Southern China, soybeans can be cultivated in multiple patterns, including double, triple, or even year-round cropping (Wang and Gai, 2002). Moreover, such cropping patterns are characterized by different intercropping and cropping rotation between soybean and other crops. Thus, how and when soybean is planted there are decided by local farmers optionally. This means that the growth phases of soybeans are inconsistent, consequently the standard curves are very hard to identify. Moreover, phenological data from local AMS could not be representative, and can't reveal local reality. The larger complexity in cropping patterns and more inputs required for

accurately identifying soybean, therefore, make us exclude the Southern China from our studied areas.


We plotted the spatial distribution of ground samples and reference points as showed by Figure 1 below and modified Figure 1 in the edited MS. We have added the details of the reference points to the data section in revised manuscript.

Reference:

[revised manuscript text omitted]

**Comment 5:** The main crop types and cropping intensity vary across regions with different climate conditions. However, Fig.3 (a-i) only presents spectral curves for soybean planting in Northern China. Are the phenological characteristics described in "(2) Feature selection" also applicable to soybeans planted in Southwestern China? I suggest that the authors also provide spectral curves of soybean and main crops planted in South China.

**Reply:** Yes, the main crop types and cropping intensity do vary across regions with different climate conditions. Using the reference points described in Reply for Comment 3, we explored the spectral and vegetation indices characteristics of major crops in each region. All these selected crops grow the similar season as those of soybeans, which further are proved by the temporal consistent profiles across different sub-zones (Fig. S2-S4). We found notable differences in SWIR1, SWIR2, and SIWSI indices between soybean and rice during the early growth period. In mid and late growth phases, EVI, NIR, Red Edge2 and Red Edge3 values of soybean fields were significantly higher than other crops. The consistent differences are basis mentioned in the feature selection section, which further substantiate that the selected features can be applicable and potentially repeatable into various regions. We have added the following figure S2-S4 to the supplementary materials and stated in the revised manuscript:

"All these spectral-phenological characteristics are also applicable to soybeans planted in other sub-zones (Fig. S2-S4)."

[Figure]

**Figure S2. Temporal profiles of (a-i) for major crops in Zone II.**

[Figure]

**Figure S3. Temporal profiles of (a-i) for major crops in Zone III.**

[Figure]

**Figure S4. Temporal profiles of (a-i) for major crops in Zone IV.**

**Comment 6:** The authors need provide example figures illustrating the result of "time window from 15 days before the podding date (DOYpodding) to 15 days after the full-seed date (DOYseed)"

**Reply:** Thank you for your insight suggestion. We have followed you to plot example figures (illustrating the result of "time window from 15 days before $DOY_{podding}$ to 15 days after $DOY_{seed}$) for identifying seasonal crops under single and double cropping patterns (Figure 2). We confined the peak value detected in soybean growing period to ensure the rationality of our method in the single or double cropping systems. We have added the figure into the revised manuscript.

[Figure]

**Figure 2. Schematic diagram of seasonal crop identification for (a) single - and (b) double - cropping systems.**

**Comment 7:** L241-242: these contents are confusing, is there any typo?

**Reply:** We are so sorry for the confusion expression. We have revised it to the follow:

"Meanwhile, the timing of TCARI reaching saturation significantly differs among soybean, rice, and wheat (Fig. 4i)."

**Comment 8:** L255-256:How did you determine the number of K-mean clusters based on statistics? Further explanation is needed for clarity.

**Reply:** We collected the statistical area for seasonal crops (including rice, maize, soybean, cotton, peanuts, sesame, sweet potato, and sorghum) of each prefecture in 2018. We defined "major crops" as those species cumulatively representing 95% of the total seasonal crop area, with an additional category for all "other crops" to determine the number of clusters $k$. We have added the process determining the number of K-mean cluster into section 2.3.2 "(3) Unsupervised learning":

"The classifier was trained individually on each prefecture based on the number of clusters $k$ input. The cluster number $k$ is defined as the number of "major crops" that constituting 95% of the total area for seasonal crops (including rice, maize, soybean, cotton, peanuts, sesame, sweet potato, and sorghum) according to prefecture-level statistics, and plus one for "other crops"."

**Comment 9:** The DTW step is not clearly described:

(1) I wonder whether the length and time coverage of S2 time series used for calculating DTW distance vary across different AEZs?

**Reply:** Yes, the length and time coverage of S2 time series is different for each sub-zone. According to the soybean sowing and maturity dates recorded at AMSs, we set the time coverage of Zone I-IV to April-September, May-October, June-October, and August-November, respectively. This selection of time spans ensures that the full growing season of soybeans is included in each sub-zone.

(2) Did the authors use averaged time series for 100 random points and those for all field samples around the whole China to calculate DTW distances? If so, it is important to note that the spectral differences between crops in North and South China may affect the validity of DTW calculation results. Have you considered the impact of intra-class spectral differences in soybean samples from different regions on the DTW calculation results and the final classification results?

**Reply:** Yes, the DTW distances are key parameters for distinguishing soybean from other crops. We determined the standard time series for each sub-zone separately. We randomly selected the 30% sample points (Dong et al., 2020) in each sub-zone and calculated the averages to determine the soybean standard curves, since the soybean growth periods and their related curves in same sub-zone do not differ hugely. For the classification results in each prefecture, we randomly select 100 points to calculate the averages and determine their standard curves for all crop category, and separately calculate the DTW distance of standard curves between the soybean and all crops.

As for your worry about the spectral differences between crops in North and South, our method proposed will not impact the DTW (calculated in a prefecture) validity because of their weak difference among a prefecture. Similarly for the intra-class spectral differences for soybean samples of different regions, such differences do not particularly impact DTW values and the final classification results because of soybean standard curves developed respectively in each sub-zone.


**Reply:** We did not use a threshold here. Based on the DTW distance of their standard curves for each crop category and soybean, the cluster with the minimum distance among all categories is selected as soybean. Taking into account of all above questions you provided, we updated the method details in the Cluster assignment section in the revised manuscript:

"We then used dynamic time warping (DTW) method to measure the similarity between each cluster's eight features involved in classification and the soybean standard curves. We averaged the data of 30% samples in each sub-zone to establish the standard curves, reducing the impact of regional phenological variations. The time coverage of Zone I-IV was set to April-September, May-October, June-October, and August-November, respectively, which are corresponding with the soybean growing season. The cluster with the minimal average DTW value was identified as the soybean cluster."

**Comment 10:** Fig.8 (a1-3) depict false-color composite images composed of bands 4, 3, and 2. Distinguishing between soybeans and non-soybeans in these images is visually difficult. It is recommended to present images composited with other bands. The authors can refer to the following article, which uses the shortwave infrared band for false-color compositing.

Song X-P, Potapov P V, Krylov A, King L, Di Bella C M, Hudson A, Khan A, Adusei B, Stehman S V,Hansen M C. National-scale soybean mapping and area estimation in the United States using medium resolution satellite imagery and field survey. Remote Sens. Environ., 2017, 190: 383-395
You N,Dong J. Examining earliest identifiable timing of crops using all available Sentinel 1/2 imagery and Google Earth Engine. ISPRS-J. Photogramm. Remote Sens., 2020, 161: 109-123

**Reply:** Thank you for your advice. We updated false color composite images (R: NIR, G: SWIR2, B: SWIR1) to identify soybean plots more clearly. The reflectance differences between soybean and other crops in these bands do be greater than that in red, green and blue bands. Many thanks for your expert advice, which really encourage us to deepen our study !

[Figure]

**Figure 3. Visual comparison of our soybean maps and existing products in typical regions in 2019.**

**Comment 11:** Fig. 9 indicates that there is a notably low frequency of clear observations in Sichuan Province, with the majority of areas showing zero clear observations per month. How can it be ensured that a complete 10-day composited time series is generated for DTW calculations in this region?

**Reply:** Yes, low frequency of clear observations was notably observed in Sichuan Province. For the areas with lower clear observations, beside the 10-day time series composite, we also conducted a gap-filling method on the composite time series by replacing the observations by the median of three adjacent observations (i.e., previous, current, and subsequent observations), to ensure the integrity of the time series as much as possible. We supplement in the "Data Processing" section:

"In areas with notably limited clear observations, a gap-filling method was conducted on the composite time series. This method involves substituting any given observation with the median value from three neighboring observations (i.e., previous, current, and subsequent observations) to maximize the continuity and completeness of time series."

Such fewer clear observations are inevitable, especially for a study over a larger region and long-term period. Although the 10-day composite time series were generated as far as possible, honestly, the uncertainty is inevitably introduced at times (such as 2017) and regions (such as the southwest)

where there are particularly few clear observations. We added discussion to the "4.2 The uncertainty from image quality" section:

"In areas with quite lower clear observations, despite a gap-filling method was conducted to generate complete 10-day composite time series, higher uncertainty is inevitable. The gap-filling time series might contain duplicate values, which cannot accurately reflect the crop growth process in reality. Obviously, the total number of images available in 2017 over the study areas was significantly fewer than those of other years (Fig.10a1-e1) ... This might explain the lower user's accuracy of soybean in Zone IV compared to other sub-zones (Table S1) and low overall accuracy based on sample verification in 2017 (Table 2)."

**Comment 12:** A considerable number of pixels corresponding to field ridges were inaccurately classified as soybeans in the 2020 map, particularly evident in East Heilongjiang, North Shandong and Henan Province. Can the authors consider the use of post-processing methods to eliminate this issue?

**Reply:** Thank you for pointing out the problem. We agree that ridge identification is a very important issue in remote sensing mapping, however it is still difficult to address the issue across a larger area. The main reasons are as follows:

(1) The ridge width is very narrow, and the 10m resolution image is often unable to accurately distinguish between the field and the ridge. It is generally accepted that the identification and elimination of the ridge is based on centimeter-level images (such as unmanned aerial vehicle images).

(2) We summarized the methods widely used to identify and eliminate the field ridges nowadays.

- Machine learning and deep learning methods. A labeled training dataset was used to train the model to identify planting areas and ridge area (Hamano et al., 2023).
- Point cloud processing technology. Point cloud data can reflect the height of the ground canopy, and the height of the ridge is often lower than that of the crop, so a suitable threshold can be adopted to distinguish the ridge from the crop (Liu et al., 2018);
- Image processing and computer vision methods. The ridge has its special shape, such as a slender shape similar to a road or a closed border. Edge detection, morphological processing and other methods can extract features from remote sensing images to help identify and distinguish ridges (Li and Qu, 2019)。

Therefore, considering the relatively weaker impacts of the field ridge on crop mapping over a larger areas, and the complex image processing algorithms (which will consume huge computing power), we have not realized the field ridge identification after trading off the cartographic accuracy and calculation cost. In future studies, with the improvement of data accuracy and algorithm update, the identification of field ridge will be a key step in large-scale crop mapping. Following your suggestions, we added the discussion to "4.3 Limitations in small-scale planting areas" section:

"Our regional adaptive large-area crop mapping method in future will further be improved by the follows: … (4) Better post-processing of data. Misclassification of field ridges and image speckles is inevitable during mapping crops over large areas. With the progress of computing power, auxiliary data and image processing algorithms can further eliminate these issues (Liu et al., 2018; Li and Qu, 2019; Hamano et al., 2023). We are sure that integrating cloud computing platforms with advanced algorithms will provide substantial potential for accurate crop identification covering larger areas in future."

---

## Author Comment (AC4)

**Response to reviewer #4:**

Many thanks for your thoughtful and valuable comments and suggestions, which are very helpful in improving our manuscript. We have conducted substantial new experiments and analyses to ensure that the study is more comprehensive and rigorous, and our maps are more reliable. Our responses to the comments point-by-point are included below in blue. The corresponding changes in the revised manuscript are shown in purple.

**General comment:** Mei et al's work mapped the soybean planting areas across China with a high spatial resolution of 10 meters, spanning from 2017 to 2021, provided important information for sustainable soybean production and management, as well as agricultural system modeling and optimization. In this work, authors summarized five methods of mapping crops by remote sensing. The advantages and uncertainties of each method were compared, and a highly effective for accurately mapping crops over a larger region method named combining unsupervised classification and post-classification methods applied in this paper. They accomplished this by Sentinel-2 remote sensing images from the GEE platform with cropland layer and detailed phenology observations. They validated the results with the census data at both county- and prefecture-level, and with the two existing datasets (CDL and GLAD maize-soybean map).

Overall, I find this work to be valuable. However, I have some concerns regarding the robustness from the sparse number of AMSs in SW Zonal IV and uncertainty in quality of satellite imagery. I hope the authors will consider these points and provide further clarification in their responses and/or revisions. Please find my major comments and minor for clarification below.

**Reply:** Thank you for your positive and constructive comments, which surely encourage us to further enhance our research quality.

To evaluate the variance in mapping accuracy across different regions, we enhanced each sub-zone's accuracy assessment using statistics and samples (see Reply for Comment 2). Zone IV's mapping results achieved a consistency $R^2$ of 0.69 with county-level statistics, deemed satisfactory (Figure S5). Validation based on samples indicated an overall soybean accuracy of 87.18% in Zone IV, though it exhibited a relatively lower producer's accuracy of 63.89% than that of other sub-zones (Table S1). Although the verification accuracy there are not as good as those in main producing areas, its accuracy is still acceptable. These findings are highlighted in our results, alongside a comparison the differences in accuracy across regions. To fully and positively respond all your valuable and suggestive comments, we also further listed them point by point in the follows.

**Major comments:**

**Comment 1:** The text mentions the need for 10-day time series composite images per month, but in certain areas, the average monthly count of clear observations is insufficient to meet this requirement. Can the existing time series composite methods be optimized to accommodate the inadequacy of observational data?

**Reply:** Yes, we have optimized the time series composite methods as possible. For the areas with lower clear observations, beside the 10-day time series composite, we also conducted a gap-filling method on the composite time series by replacing the observations by the median of three adjacent observations (i.e., previous, current, and subsequent observations), to ensure the integrity of the time series as much as possible. We supplement in the "Data Processing" section:

"In areas with notably limited clear observations, a gap-filling method was conducted on the composite time series. This method involves substituting any given observation with the median value from three neighboring observations (i.e., previous, current, and subsequent observations) to maximize the continuity and completeness of time series."

Naturally, although the 10-day composite time series were generated as far as possible, this inevitably introduces uncertainty at times (such as 2017) and regions (such as the southwest) where there are particularly few clear observations. We added discussion to the "4.2 The uncertainty from image quality" section:

"In areas with quite lower clear observations, despite a gap-filling method was conducted to generate complete 10-day composite time series, higher uncertainty is inevitable. The gap-filling time series might contain duplicate values, which cannot accurately reflect the crop growth process in reality. Obviously, the total number of images available in 2017 over the study areas was significantly fewer than those of other years (Fig.10a1-e1) ... This might explain the lower user's accuracy of soybean in Zone IV compared to other sub-zones (Table S1) and low overall accuracy based on sample verification in 2017 (Table 2)."

**Comment 2:** The observations per month of satellite imagery in SW Zonal IV are less, and the AMSs in this zonal also only have two sites. Whether it is possible to increase the observational data or phenological data from remote sensing to test the robust.

**Reply:** Your suggestion is very helpful. In order to test the robustness of mapping in different regions, we supplemented the statistical data validation of partitions and the point validation based on existing data sets, further demonstrating the mapping accuracy of Zone IV. Zone IV's mapping results achieved a consistency $R^2$ of 0.69 with county-level statistics, deemed satisfactory (Figure S5). Validation based on samples indicated an overall soybean accuracy of 0.87 in Zone IV, though it exhibited a relatively lower producer accuracy of 0.64 than that of other sub-zones (Table S1). Overall, the accuracy of each sub-zone is acceptable despite some variations.

➢ **The variations in accuracy among sub-zones based on statistics validation:**

"The mapping accuracy in Zone I closely matched county-level statistics, showing high consistency ($R^2$=0.86). Zones II-IV also demonstrated reasonable agreement ($R^2$=0.50~0.69), despite relatively lower accuracy due to the scarcer planted areas (Fig. S5). No significant trend deviation from statistics was indicated for the mapping area in Zone I, with slight overestimates for Zone II and III, and underestimates for Zone IV (Fig. S5). These accuracy variations are acceptable, given the challenges in accurately identifying soybeans in regions where they are planted less prevalently. Specifically, maize is more dominant than soybeans in Zone II, while Zone III is characterized by diverse crops and complex planting patterns. Underestimation in Zone IV is possibly due to fewer clear observations in the southwest. Nevertheless, the overall accuracy across the zones is acceptable."

[Figure]

**Figure S5. Comparison of soybean areas with county-level statistics in (a) Zone I, (b) Zone II, (c) Zone III, and (d) Zone IV in 2017 and 2018.**

➢ **The variations in accuracy among sub-zones based on samples validation:**

"The overall accuracy for each sub-zone in 2019 varied from 83.58% to 90.67% (Table S1). Specifically, Zone I demonstrated the highest producer's accuracy for soybean at 88.31%, aligning with its high consistency with statistics. Zone III achieved the highest overall accuracy at 90.67%, attributed to its superior user's accuracy for soybean, indicating fewer misclassifications, and effective differentiation from non-soybean crops (Table S1). The producer's accuracy in Zone IV was relatively lower at 63.89%, possibly due to the limited samples, high heterogeneity, and fewer clear observations (Table S1)."

**Table S1. Confusion matrix of the soybean maps in each sub-zone in 2019.**

|  | Reference | Map | | Producer's | User's | F1 | Overall |
|---|---|---|---|---|---|---|---|
|  |  | Soybean | Non-Soybean | Accuracy | Accuracy | Score | Accuracy |
| I | Soybean | 922 | 122 | 88.31% | 81.09% | 0.85 | 87.12% |
|  | Non-Soybean | 215 | 1358 | 86.33% | 91.76% | 0.89 |  |
| II | Soybean | 233 | 74 | 75.90% | 86.30% | 0.81 | 83.58% |
|  | Non-Soybean | 37 | 332 | 89.97% | 81.77% | 0.86 |  |
| III | Soybean | 101 | 26 | 79.53% | 98.06% | 0.88 | 90.67% |
|  | Non-Soybean | 2 | 171 | 98.84% | 86.80% | 0.92 |  |
| IV | Soybean | 23 | 13 | 63.89% | 92.00% | 0.75 | 87.18% |
|  | Non-Soybean | 2 | 79 | 97.53% | 85.87% | 0.91 |  |

**Comment 3:** To determine the potential cropping areas, authors filtered the pixels exhibiting an EVI maximum value during the growing season greater than 0.4 to remove fallow land. For spatial variation across four zonal, the constant threshold would bring some uncertainty. I expect to see more evidence for selecting 0.4 or a sensitivity analysis of threshold can also be implemented.

**Reply:** We identified the pixels with maximum EVI values < 0.4 as fallow land because the maximum EVI values for crops are all > 0.4 (except for few outliers) based on all ground samples in 2019 (Figure S1). In addition, studies on crop mapping across China also put forward that EVI values in croplands generally exceed 0.4 at peak growth (Li et al., 2014; Zhang et al., 2017; Han et al., 2022). Thus, using 0.4 as a threshold allows us to strictly remove fallow land. We have provided additional explanations for the threshold choice in the revised manuscript:

"Based on the cropland extracted, we filtered out the pixels exhibiting an Enhanced Vegetation Index (EVI) maximum value during the growing season less than 0.4 to remove fallow land according to the analysis of ground samples (Fig. S1) and previous studies, which found that almost all crops had maximum EVI values above 0.4 (Li et al., 2014; Zhang et al., 2017; Han et al., 2022)."

Reference:

Han, J., Zhang, Z., Luo, Y., Cao, J., Zhang, L., Zhuang, H., Cheng, F., Zhang, J., and Tao, F.: Annual paddy rice planting area and cropping intensity datasets and their dynamics in the Asian monsoon region from 2000 to 2020, Agric. Syst., 200, 103437, https://doi.org/10.1016/j.agsy.2022.103437, 2022.

Li, L., Friedl, M. A., Xin, Q., Gray, J., Pan, Y., and Frolking, S.: Mapping Crop Cycles in China Using MODIS-EVI Time Series, Remote Sens., 6, 2473–2493, https://doi.org/10.3390/rs6032473, 2014.

Zhang, G., Xiao, X., Biradar, C. M., Dong, J., Qin, Y., Menarguez, M. A., Zhou, Y., Zhang, Y., Jin, C., Wang, J., Doughty, R. B., Ding, M., and Moore, B.: Spatiotemporal patterns of paddy rice croplands in China and India from 2000 to 2015, Sci. Total Environ., 579, 82–92, https://doi.org/10.1016/j.scitotenv.2016.10.223, 2017.

[Figure]

**Figure S1. Box plot of the EVI maximum in 2019 based on all ground samples.**

**Minor comments:**

**Comment 4:** Line 58: "same areas" means the north China?

**Reply:** Yes, "same areas" here refers to northeast China. We changed it to "three provinces of Northeast China" in the revised manuscript to clarify the meaning.

**Comment 5:** Line 180, Figure2: The label on the left in Figure2 (i.e. 'Data processing' and 'Accuracy assessment') are set to rotate 180° to match reading habits.
**Reply:** Thank you for your suggestion. We have angled all the labels on the left side of the diagram for easier reading (Figure 2).

**Comment 6:** Line 180, Figure2: In step2, part (2) of the dashed box is confusing. What the color represents? If I understand correctly, they represent different layers of indexes. It is recommended to put the abbreviation to the right of the color layers.
**Reply:** Yes, we have redrawn part (2) of the figure and marked the band or index abbreviations accordingly as you suggested (Figure 2).

[Figure]

**Figure 2. The Regional Adaption Spectra-Phenology Integration methodology for retrieving soybean planting area.**

---

## Author Response (AR2)

Dear Editors and Reviewers:

Sincere thanks for the evaluation of this work again and your valuable comments and suggestions for improving this manuscript. We carefully considered the suggestions and made some changes to the revised manuscript (essd-2023-467). Here we submit the revised version, which has been modified according to the comments from the reviewers.

We attach the detailed item-by-item response to all comments and suggestions for the evaluation. Our responses to the comments point-by-point are included below in blue. The corresponding changes in the revised manuscript are shown in purple.

Yours sincerely,

Zhao Zhang and co-authors
* * *
**Reviewer #1:**

I appreciate the authors' efforts in addressing my previous comments and those from the other reviewers. The manuscript has shown improvement in presentation. I have one additional comment regarding the revised manuscript: the added section titled "4.1 Our Advantages and Potential Applicability" might make more sense if integrated with the analysis comparing different data products/methods, rather than as a standalone section.

**Response:** Thank you for your suggestion. We integrated with the analysis comparing with others for both products and methods (**Lines 420-444**). We have added a comparison of the ChinaSoyArea10m with other products in Section 4.1. Additionally, on the basis of the advantages of the methodology, we combined the comparison with other commonly used methods to make the content of this section more complete. It now read as:

**4.1 Our advantages and potential applicability**

We proposed a new framework (RASP) to identify annual dynamic of soybean planting areas over larger regions and produced the longer-term series of soybean maps (ChinaSoyArea10m) across mainly planting areas in China from 2017 to 2021 at the first time. The accuracy of ChinaSoyArea10m is acceptable ($R^2 \sim 0.85$) at both county- and prefecture-level, with relatively less $R^2$ than GLAD ($R^2 = 0.93$ at prefecture-level), but higher than CDL ($R^2 = 0.53$ at county-level). Compared with existing products, ChinaSoyArea10m accurately depict the soybean with more spatial and temporal details as well.

The methodology developed for identifying soybean planting areas indicate several notable strengths that make it an attractive option for wide application. Firstly, it operates independently, without extensive ground samples required. The conventional supervised approaches like random forest (RF) and long short-term memory (LSTM) depend on quantities of observations, with much money, time, and labor consumed. In this context, both transferable learning model and our RASP methods (combing unsupervised learning with statistics) indeed provide huge potential for crop mapping. However, transferable models are suitable for areas or years with similar cropping patterns. In areas with diverse and complex cropping patterns, it is a challenge to apply the supervised model trained in limited areas or limited years into others (Wang et al., 2019; Ma et al., 2020). In contrast, our strategy leverages a specific, pre-existing set of samples to stably differentiate soybean characteristics from other crops, which can accurately map annual dynamics without updated requirement in annual samples. Consequently, this method significantly weakens limitations in crop classification during years without specific samples, enabling crop mapping consistently and

continually.

Another key advantage of our spectra-phenology integration approach is its quick applicability over larger areas, coupled with excellent spatial scalability. It can self-adopt to different environments by considering phenology information. Compared to methods that rely on composite indicators and specific thresholds, our approach simplifies the requirements for inputs and experienced judgements. The only inputs required are the phenological information of soybeans and the number of other primary crops during the same growing season in the targeted area. This allows to classify crop swiftly and efficiently without additional inputs for background knowledge or setting complex thresholds. The input of phenological information in each prefecture enhanced the zonal adaptive assessment of soybean growth status across various areas, thereby facilitating crop classification. This innovative approach ensures its applicability into other soybean-producing areas, showcasing its potential for broader implementation.

**Reviewer #2:**

I have just a few minor comments and suggestions on the revised manuscript.

(1) Why is the total number of Sentinel-2 observations and clear observations in 2017 much lower than in the other four years? It looks strange because the whole country had much lower data availability.

**Response:** Thank you for your question. The fewer observations in 2017 was due to Sentinel-2B launched in March of that year. Prior to March, only Sentinel-2A was operational, resulting in a 10-day revisit period. Once Sentinel-2B was stably operational and began providing data, the revisit frequency improved to every five days in many areas, increasing the number of observations. We have supplemented the reasons for data availability in Section 4.2 of revised MS (Lines **465-466**).

"Obviously, the total number of images available in 2017 over the study areas was significantly fewer than those of other years, because the second satellite Sentinel-2B only commenced operations and started providing data after March of 2017 (Fig.10a1-e1)."

(2) Unsupervised classification and cluster assignment is still not very clear to me. Line 296-297, What do you mean by 'using key phenological information as input features'? My understanding is that phenological information is not directly used as features for classification. Also, six bands and two indices were used for classification. When DTW is used, do you calculate the DTW values between each indicator with the corresponding soybean curve and average the eight values? Or do you simply make the decision based on the minimum DTW values among the eight indicators? Need more clarification.

**Response:** Thank you for your suggestions.

For Q1, we are sorry for the confusion expression. We identified key soybean growth stages (VGP, RGP) using phenological records and selected the spectral features and vegetation indices from these periods as input features. Specifically, within the VGP, we used 2 SWIR bands and the SIWSI index, and within the RGP, we used 3 RE bands, 2 SWIR bands, NIR bands, and the TCARI index. All there were clearly showed in Figure 2 of the MS. We have updated the descriptions in the MS accordingly (**Lines 296-297**).

"We used the detailed phenological records at AMSs to identify soybean growth periods and selected the spectra and vegetation indices within specific growth periods (VGP, RGP) key phenological information

as input features."

For Q2, we calculated the DTW distances for these 8 features and used the averaged DTW distance for all features to assess the similarity degree with the standard curve. Based on the DTW distance of their standard curves for each crop category and soybean, the cluster with the minimum distance among all categories is selected as soybean. We have clarified the details in the Cluster assignment section in the MS (**Lines 305-311**):

"We then used dynamic time warping (DTW) method to measure the similarity between each cluster's eight features involved in classification and the soybean standard curves. We averaged the data of 30% samples in each sub-zone to establish the standard curves, reducing the impact of regional phenological variations. The time coverage of Zone I-IV was set to April-September, May-October, June-October, and August-November, respectively, which are corresponding with the soybean growing season. The cluster with the minimal average of 8 DTW values was identified as the soybean cluster."

(3) Considering the different cropping patterns of the major crops in each zone, it is better to add the time series profiles of major crops at different zones to present the separatability of the clusters. Could be included in the supplementary documents.

**Response:** Thank for your suggestion. Indeed, cropping patterns vary across regions. We analyzed the spectral and vegetation indices characteristics of major crops in each zone, selecting those that grow in the same season as soybeans (Fig. S2-S4). We observed distinct differences in the SWIR1, SWIR2, and SIWSI indices between soybean and rice during their early growth stages. In the mid and late growth phases, the EVI, NIR, Red Edge2, and Red Edge3 values were significantly higher in soybean compared to other crops. These consistent differences support the applicability and potential repeatability of the selected features across various regions. Figures S2-S4 have been added to the supplementary materials and referenced in the revised manuscript (**Lines 281-282**).

"All these spectral-phenological characteristics are also applicable to soybeans planted in other sub-zones (Fig. S2-S4)."

[Figure]

**Figure S2. Temporal profiles of (a-i) for major crops in Zone II.**

[Figure]

**Figure S3. Temporal profiles of (a-i) for major crops in Zone III.**

[Figure]

**Figure S4. Temporal profiles of (a-i) for major crops in Zone IV.**